



Ocean biogeochemistry in the Canadian Earth System Model version 5.0.3: CanESM5 and
CanESM5-CanOE
James R. Christian[1,2], Kenneth L. Denman[2,3], Hakase Hayashida[3,4], Amber M. Holdsworth[1],
Warren G. Lee[2], Olivier G.J. Riche[3,5], Andrew E. Shao[2,3], Nadja Steiner[1,2], and Neil C. Swart[2]
1 Fisheries and Oceans Canada, Sidney, BC, Canada
2 Canadian Centre for Climate Modelling and Analysis, Victoria, BC, Canada
3 School of Earth and Ocean Sciences, University of Victoria, Victoria, BC, Canada
4 now at the Institute for Marine and Antarctic Studies, University of Tasmania, Hobart,
Tasmania, Australia
5 now at Fisheries and Oceans Canada, Mont Joli, Québec, Canada
*Correspondence to*: James Christian (jim.christian@ec.gc.ca)





**Abstract.** The ocean biogeochemistry components of the Canadian Earth System Model v. 5 are
presented and compared to observations and other models. CanESM5 employs the same
biogeochemistry module as CanESM2 whereas CanESM5-CanOE ("Canadian Ocean Ecosystem
model") is a new, more complex biogeochemistry module developed for CMIP6, with multiple
food chains, flexible phytoplankton elemental ratios, and a prognostic iron cycle. This new
model is described in detail and the outputs compared to CanESM5 and CanESM2, as well as to
observations and other CMIP6 models. Both CanESM5 models show gains in skill relative to
CanESM2, which are attributed primarily to improvements in ocean circulation. CanESM5-
CanOE shows improved skill relative to CanESM5 in some areas. CanESM5-CanOE includes a
prognostic iron cycle, and maintains high nutrient / low chlorophyll conditions in the expected
regions (in CanESM2 and CanESM5, iron limitation is specified as a temporally static 'mask').
Surface nitrate concentrations are biased low in the subarctic Pacific and equatorial Pacific, and
high in the Southern Ocean. Export production in CanESM5-CanOE is among the lowest for
CMIP6 models; in CanESM5 it is among the highest, but shows the most rapid decline after
about 1980. CanESM5-CanOE has relatively low concentrations of zooplankton and detritus
relative to phytoplankton, and a high and relatively constant living phytoplankton fraction of
total particulate organic matter. In most regions, large and small phytoplankton show decoupled
seasonal cycles with greater abundance of large phytoplankton in the productive seasons.
Cumulative ocean uptake of anthropogenic carbon dioxide through 2014 is lower in both
CanESM5 models than in observation-based estimates or the model ensemble mean, and is lower
in CanESM5-CanOE (122 PgC) than in CanESM5 (132 PgC).



## 1. **Introduction**

The Canadian Centre for Climate Modelling and Analysis has been developing coupled models with an interactive carbon cycle for more than a decade (Arora et al., 2009; 2011; Christian et al., 2010). CanESM5 (Swart et al., 2019a) is an updated version of CanESM2 (Arora et al., 2011), with an entirely new ocean and an atmosphere with the same T63 horizontal resolution and important improvements in atmospheric physics. The CanESM5 ocean is based on the Nucleus for European Modelling of the Ocean (NEMO) system version 3.4. The ocean biogeochemistry modules were developed in-house, although parameterizations for some processes were adapted from the native PISCES biogeochemistry model (Aumont et al., 2015). CanESM5 uses the same biogeochemistry model as CanESM1 and CanESM2, the Canadian Model of Ocean Carbon (CMOC; Zahariev et al., 2008), adapted for use within NEMO. An additional model was developed for CMIP6, called the Canadian Ocean Ecosystem model (CanOE). The biological components of CanOE are of substantially greater complexity, including multiple food chains, flexible phytoplankton elemental ratios, and a prognostic iron (Fe) cycle. Carbon chemistry, gas exchange and solubility of carbon dioxide ($CO_2$) and oxygen are identical between the two and follow the protocols specified by the Ocean Model Intercomparison Project - Biogeochemistry (OMIP-BGC) (Orr et al., 2017). The two coupled models are known as CanESM5 and CanESM5-CanOE, respectively. There are no feedbacks between biology and the physical ocean model, so the physical climate of CanESM5 and CanESM5-CanOE is identical in experiments with prescribed atmospheric $CO_2$ concentration.

The reasons for developing both models are, firstly, to evaluate the effect of changes in ocean



circulation between CanESM2 and CanESM5 on ocean biogeochemistry by running the new
climate model with the same ocean biogeochemistry, and secondly because CanOE is
substantially more expensive computationally (19 tracers vs 7). Having CMOC as an option
allowed us to run many CMIP6 experiments with CanESM5 only, as ocean biogeochemistry is
not central to their purpose. Many additional tracers requested by OMIP-BGC including abiotic
and natural dissolved inorganic carbon (DIC), $DI^{14}C$, CFCs and $SF_6$ (see Orr et al.,2017) were
run only in CanESM5, since adding these tracers on top of the larger set of biological
components in CanOE would have been prohibitively expensive. The CMIP6 experiments
published for CanESM5-CanOE are listed in Supplementary Table S1.

CMOC is a nutrient-phytoplankton-zooplankton-detritus (NPZD) model with highly
parameterized representations of phytoplankton Fe limitation, dinitrogen ($N_2$) fixation and
denitrification, and calcification and calcite dissolution (Zahariev et al., 2008). In CanESM1 and
CanESM2, CMOC did not include oxygen. In CanESM5, CMOC now includes oxygen as a
purely 'downstream' tracer that does not affect other biogeochemical processes, whereas in
CanOE denitrification is prognostic and dependent on the concentration of oxygen. Among the
less satisfactory aspects of CMOC biogeochemistry are, firstly, that Fe limitation is specified as a
static 'mask' that does not change with climate (being calculated from the present-day
climatological distribution of nitrate), and secondly, that denitrification is parameterized so that
nitrogen (N) is conserved within each vertical column, i.e., collocated with $N_2$ fixation in tropical
and subtropical open-ocean regions (Zahariev et al., 2008; Riche and Christian, 2018). This latter
simplification produced excessive accumulations of nitrate in Eastern Boundary Current regions
where most denitrification actually occurs. CMOC also has a tendency to produce rather stark





extremes of high and low primary and export production (Zahariev et al., 2008), a well-known
problem of NPZD models (Armstrong, 1994; Friedrichs et al., 2007). Our intent in developing
CanOE was to alleviate, or at least reduce, these biases, by including multiple food chains, a
prognostic Fe cycle, and prognostic denitrification. Dinitrogen fixation is still parameterized, but
the CanOE parameterization includes Fe limitation, whereas in CMOC $N_2$ fixation tends to grow
without bound in a warming ocean as there is no P or Fe limitation (Riche and Christian, 2018).
Calcification is represented by a prognostic detrital calcite pool with its own sinking rate (distinct
from that of organic detritus), and calcite burial depends on the saturation state. In CMOC
calcification is parameterized by a temperature dependent "rain ratio" and 100% burial of calcite
that reaches the seafloor is assumed.

In this paper we present a detailed model description for CanOE and an evaluation of both
CanESM5 and CanESM5-CanOE relative to observational data products and other available
models. CMOC has been well described previously (Zahariev et al., 2008) and the details are not
reiterated here. In some cases, CanESM2 results are also shown to illustrate which differences in
the model solutions arise largely from the evolution of the physical ocean model, and which are
specifically associated with different representations of biogeochemistry.

**2. Model Description**

The NEMO system is a publicly available archive of codes based on the OPA (Océan
PArallelisé) ocean model (Madec and Imbard, 1996; Guilyardi and Madec, 1997). It comes with
two options for biogeochemistry: PISCES (Pelagic Interactions Scheme for Carbon and



Ecosystem Studies) and LOBSTER (LODyC Ocean Biogeochemical System for Ecosystem and
Resources). CanOE and the NEMO implementation of CMOC are built around the basic code
structure of PISCES within the Tracers in Ocean Paradigm (TOP) module, using NEMO v3.4.1,
but have also been implemented in NEMO 3.6 for regional downscaling applications
(Holdsworth et al., 2021).

The biology, carbon chemistry, gas exchange and light attenuation components have all been
modified to various degrees. In a few cases PISCES parameterizations, or slightly modified
versions thereof, were adopted. CanOE uses PISCES three-band light attenuation while NEMO-
CMOC uses broadband attenuation of photosynthetically active radiation (PAR) for consistency
with the published version of CMOC. Carbon chemistry was modified to be consistent with the
Best Practices Guide (Dickson et al., 2007) and the OMIP-BGC data request (Orr et al., 2017).
All calculations are done on the total scale and the recommended formulae for the equilibrium
constants are employed. The PISCES conventions for convergence of carbon chemistry
calculations were retained, the greater number of iterations in the surface layer offering greater
accuracy in calculating $pCO_2$ and gas exchange. In subsurface layers (where the only function of
the carbon chemistry is to calculate burial of calcite in the sediments) the number of iterations is
fixed at five. CanESM5 uses the same carbon chemistry but does not solve the carbon chemistry
equations in the subsurface layers.

The CanOE biology model is a substantially new model based on the cellular regulation model
of Geider et al. (1998). There are two phytoplankton functional groups, and each group has four
state variables: C, N, Fe and chlorophyll. Photosynthesis is decoupled from cell production and





photosynthetic rate is a function of the cell's internal N and Fe quotas. Each functional group has
a specified minimum and maximum N quota and Fe quota, and nutrient uptake ceases when the
maximal cell quota is reached. Chlorophyll synthesis is a function of N uptake and increases at
low irradiance. Model parameters and their values are listed in Table 1. A schematic of the
model is shown in Figure 1.

**2.1 Photosynthesis and Phytoplankton Growth**

For simplicity and clarity, the equations are shown here for a single phytoplankton species, and
do not differ structurally for small and large phytoplankton. Some parameter values differ for the
two phytoplankton groups; all parameter values are listed in Table 1.

Temperature dependence of photosynthetic activity is expressed by the Arrhenius equation
$$T_f = \exp\left(-\frac{E_{ap}}{R}\left(\frac{1}{T} - \frac{1}{T_{\text{ref}}}\right)\right) \qquad (1)$$
where $E_{ap}$ is an enzyme activation energy that corresponds approximately to that of RuBisCo (cf.
Raven and Geider 1988), $R$ is the gas constant (8.314 J mol$^{-1}$ K$^{-1}$), and temperature $T$ and
reference temperature $T_{ref}$ are in Kelvin. Maximal rates of nutrient (either N or Fe, but
generically referred to here with the superscript X) uptake are given by
$$V_{max}^X = V_{ref}^X T_f \left(\frac{Q_{max}^X - Q^X}{Q_{max}^X - Q_{min}^X}\right)^{0.05} \qquad (2)$$
where $V_{max}{}^X$ is the maximal uptake rate in mg of nutrient X per mg of cell C, X can represent N
or Fe, $Q$ is the nutrient cell quota and $Q_{min}$ and $Q_{max}$ its minimum and maximum values, and $V_{ref}{}^X$





Figure 1 - Schematic of the CanOE biology model. Model currencies including chlorophyll (Chl) are indicated by coloured boxes except oxygen ($O_2$) and carbonate ($CaCO_3$). Arrows indicate flows of carbon (C), nitrogen (N) and iron (Fe) between compartments containing small (S) and large (L) phytoplankton (P), zooplankton (Z), and detritus (D) components; counterflows of oxygen are not shown.





Table 1 – Ecosystem model parameters.

| Symbol | Description | Unit | |
|---|---|---|---|
| | | | |
| $T_{ref}$ | Reference temperature | K | 298.15 |
| $E_{ap}$ | Activation energy for photosynthesis | kJ mol$^{-1}$ | 37.4 |
| $Q^N_{mins}$ | Small phytoplankton minimum N quota | g N g C$^{-1}$ | 0.04 |
| $Q^N_{maxs}$ | Small phytoplankton maximum N quota | g N g C$^{-1}$ | 0.172 |
| $Q^N_{minl}$ | Large phytoplankton minimum N quota | g N g C$^{-1}$ | 0.04 |
| $Q^N_{maxl}$ | Large phytoplankton maximum N quota | g N g C$^{-1}$ | 0.172 |
| $Q^{Fe}_{mins}$ | Small phytoplankton minimum Fe quota | µg Fe g C$^{-1}$ | 4.65 |
| $Q^{Fe}_{maxs}$ | Small phytoplankton maximum Fe quota | µg Fe g C$^{-1}$ | 93. |
| $Q^{Fe}_{minl}$ | Large phytoplankton minimum Fe quota | µg Fe g C$^{-1}$ | 6.5 |
| $Q^{Fe}_{maxl}$ | Large phytoplankton maximum Fe quota | µg Fe g C$^{-1}$ | 70. |
| $V^N_{ref}$ | Reference rate of N uptake | g N g C$^{-1}$ d$^{-1}$ | 0.6 |
| $V^{Fe}_{ref}$ | Reference rate of Fe uptake | µg Fe g C$^{-1}$ d$^{-1}$ | 79. |
| $P^C_{ref}$ | Reference rate of photosynthesis | g C g C$^{-1}$ d$^{-1}$ | 3 |
| $k_{XU}$ | Rate coefficient for exhudation | d$^{-1}$ | 1.7 |
| $k_{dgr}$ | Rate coefficient for chlorophyll degradation | d$^{-1}$ | 0.02 |
| $\zeta$ | Respiratory cost of biosynthesis | g C g N$^{-1}$ | 2 |
| $\alpha_{chl}$ | Initial slope of P-E curve | g C g CHL$^{-1}$ h$^{-1}$ (µmol m$^{-2}$ s$^{-1}$)$^{-1}$ | 1.08 |
| $\Theta^N_{max}$ | Maximum chlorophyll-nitrogen ratio | g g$^{-1}$ | 0.18 |
| $K_{NiS}$ | Half-saturation for small phytoplankton nitrate uptake | mmol$^{-1}$ m$^3$ | 0.1 |
| $K_{NaS}$ | Half-saturation for small phytoplankton ammonium uptake | mmol$^{-1}$ m$^3$ | 0.05 |
| $K_{FeS}$ | Half-saturation for small phytoplankton iron uptake | nmol$^{-1}$ m$^3$ | 100 |
| $K_{NiL}$ | Half-saturation for large phytoplankton nitrate uptake | mmol$^{-1}$ m$^3$ | 1.0 |
| $K_{NaL}$ | Half-saturation for large phytoplankton ammonium uptake | mmol$^{-1}$ m$^3$ | 0.05 |
| $K_{FeL}$ | Half-saturation for large phytoplankton iron uptake | nmol$^{-1}$ m$^3$ | 200 |
| $m_{1S}$ | Small phytoplankton/zooplankton mortality rate (linear) | d$^{-1}$ | 0.05 |
| $m_{2S}$ | Small phytoplankton/zooplankton mortality coefficient | (mmol C m$^{-3}$)$^{-1}$ d$^{-1}$ | 0.06 |
| $m_{1L}$ | Large phytoplankton/zooplankton mortality rate (linear) | d$^{-1}$ | 0.1 |





| $m_{2L}$ | Large phytoplankton/zooplankton mortality coefficient | (mmol C $m^{-3}$)$^{-1}$ d$^{-1}$ | 0.06 |
|---|---|---|---|
| $X_{minp}$ | Minimum phytoplankton concentration for linear mortality | mmol C $m^{-3}$ | 0.01 |
| $a_L$ | Large zooplankton grazing parameter | (mmol C $m^{-3}$)$^{-1}$ | 0.25 |
| $G_{L0}$ | Large zooplankton maximum grazing rate | d$^{-1}$ | 0.85 |
| $a_S$ | Small zooplankton grazing parameter | (mmol C $m^{-3}$)$^{-1}$ | 0.25 |
| $G_{S0}$ | Small zooplankton maximum grazing rate | d$^{-1}$ | 1.7 |
| $\lambda$ | Assimilation efficiency | n.d. | 0.8 |
| $r_{zs}$ | Microzooplankton specific respiration rate at $T_{ref}$ | d$^{-1}$ | 0.3 |
| $r_{zl}$ | Mesozooplankton specific respiration rate at $T_{ref}$ | d$^{-1}$ | 0.1 |
| $r_1$ | Small detritus remineralization rate at $T_{ref}$ | d$^{-1}$ | 0.25 |
| $r_2$ | Large detritus remineralization rate at $T_{ref}$ | d$^{-1}$ | 0.25 |
| $E_{ar}$ | Activation energy for detritus remineralization | kJ mol$^{-1}$ | 54.0 |
| $w_s$ | Small detritus sinking speed | m d$^{-1}$ | 2. |
| $w_l$ | Large detritus sinking speed | m d$^{-1}$ | 30. |
| $w_{Ca}$ | $CaCO_3$ sinking speed | m d$^{-1}$ | 20. |
| $P_{Ca}$ | $CaCO_3$ production as fraction of mortality | mol $CaCO_3$ molC$^{-1}$ | 0.05 |
| $k_{Ca}$ | $CaCO_3$ dissolution rate | d$^{-1}$ | 0.0074 |
| $S_{Fe1}$ | Dissolved iron scavenging loss rate (Fe$\leq$L$_{Fe}$) | d$^{-1}$ | 0.001 |
| $S_{Fe2}$ | Dissolved iron scavenging loss rate (Fe>L$_{Fe}$) | d$^{-1}$ | 2.5 |
| $L_{Fe}$ | Ligand concentration | nmol Fe $m^{-3}$ | 600. |
| $P_{Fe}$ | POC-dependence parameter for Fe scavenging | (mmolC $m^{-3}$)$^{-1}$ | 0.66 |
| $K_{NH4ox}$ | Nitrification rate in darkness | d$^{-1}$ | 0.05 |
| $K_E$ | Half-saturation for irradiance inhibition of nitrification | W $m^{-2}$ | 1. |
| $k_{dnf}$ | Light and nutrient saturated rate of $N_2$ fixation at 30°C | mmol $m^{-3}$ d$^{-1}$ | 0.0225 |
| $a$ | Initial slope for irradiance-dependence of $N_2$ fixation | (W $m^{-2}$)$^{-1}$ | 0.02 |
| $K_{Fe}$ | Half-saturation for Fe dependence of $N_2$ fixation | nmol $m^{-3}$ | 100. |
| $K_{NO3}$ | Half-saturation for DIN inhibition of $N_2$ fixation | mmol $m^{-3}$ | 0.1 |
| $O_{mxd}$ | $O_2$ concentration threshold for denitrification | mmol $m^{-3}$ | 6. |
| $A_f$ | Anammox fraction of N loss to denitrification | n.d. | 0.25 |



is a (specified) basal rate at $T=T_{ref}$ and $Q=Q_{min}$. These maximum rates are then reduced according
to the ambient nutrient concentration, i.e.
$$V^N = V_{max}^N (L_{NH4} + (1 - L_{NH4}) L_{NO3}) \tag{3a}$$
where $L_{\mathrm{NH4}} = \frac{N_a}{K_{\mathrm{NaX}} + N_a}$ and $L_{\mathrm{NO3}} = \frac{N_i}{K_{\mathrm{NiX}} + N_i}$ , with $N_i$ and $N_a$ indicating nitrate and ammonium
respectively, and
$$V^{Fe} = V_{max}^{Fe} \left( \frac{\mathrm{Fe}}{K_{\mathrm{FeX}} + \mathrm{Fe}} \right) \tag{3b}$$
where X indicates large or small phytoplankton (Table 1). The maximal carbon-based growth
rate is given by
$$P_{max}^C = P_{ref}^C T_f \min\left\{ \frac{Q^N - Q_{min}^N}{Q_{max}^N - Q_{min}^N} \cdot \frac{Q^{Fe} - Q_{min}^{Fe}}{Q_{max}^{Fe} - Q_{min}^{Fe}} \right\} \tag{4}$$
where $P^C_{ref}$ is the rate at the reference temperature $T_{ref}$ under nutrient-replete conditions
($Q=Q_{max}$). The light-limited growth rate is then given by
$$P_{phot}^C = P_{max}^C \left( 1 - e^{-\alpha_{chl} E \theta_C / P_{max}^C} \right) \tag{5}$$
where $\theta_C$ is the chlorophyll-to-carbon ratio. The rate of chlorophyll synthesis is
$$\rho_{\mathrm{chl}} = \theta_{max}^N \frac{P_{phot}^C}{E \alpha_{\mathrm{chl}} \theta} \tag{6}$$
These rates are then used to define a set of state equations for phytoplankton carbon ($C_p$),
nitrogen ($N_P$), iron ($Fe_P$), and chlorophyll (M).
$$\frac{dC_p}{dt} = (P_{phot}^C - \zeta V_N) C_p - (G + C_{XS}) - m_1 C_p - m_2 C_p^2 - k_{XU} C_{INTR} \tag{7}$$
where $\zeta$ is the respiratory cost of biosynthesis, $G$ is the grazing rate (equation 12), $C_{XS}$ is the
excess (above the ratio in grazer biomass) carbon in grazing losses, $m_1$ and $m_2$ are coefficients
for linear and quadratic nonspecific mortality terms, $C_{INTR}$ is the concentration of intracellular



carbohydrate carbon in excess of biosynthetic requirements, and $k_{XU}$ is a rate coefficient for its
exudation to the environment. The nonspecific mortality terms are set to 0 below 0.01 mmol C
$m^{-3}$, to prevent biomass from being driven to excessively low levels in the high latitudes in
winter (Hayashida, 2018). The full equation for phytoplankton N, Fe and chlorophyll are
$$\frac{dN_p}{dt} = \frac{V^N}{Q_N} - (G + m_1 C_p + m_2 C_p^2)R_{NC} - N_{XS} \tag{8}$$

$$\frac{dFe_p}{dt} = \frac{V^{Fe}}{Q_{Fe}} - (G + m_1 C_P + m_2 C_p^2)R_{FeC} - Fe_{XS} \tag{9}$$

$$\frac{dM}{dt} = \frac{\rho_{chl}V^N}{\theta_C}M - (G + m_1 C_p + m_2 C_p^2)\theta_C - k_{dgr}M \tag{10}$$

where $k_{dgr}$ is a rate coefficient for nonspecific losses of chlorophyll e.g., by photooxidation, in
addition to losses to grazing and other processes that also affect $C_p$, $N_p$, and $Fe_p$. $N_{XS}$ and $Fe_{XS}$ are
remineralization of "excess" (relative to grazer or detritus ratios) N or Fe and are defined below
(equation 16).

**2.2 Grazing and Food Web Interactions**

Grazing rate depends on the phytoplankton carbon concentration, which most closely represents
the food concentration available to the grazer (Elser and Urabe 1999; Loladze et al. 2000).
Zooplankton biomass is also in carbon units. State equations for small and large zooplankton are
$$\frac{dZ_s}{dt} = \lambda G_s - (R + G_Z + m_{1s}Z_s + m_{2s}Z_s^2) \tag{11a}$$

$$\frac{dZ_L}{dt} = \lambda G_L - (R + m_{1L}Z_L + m_{2L}Z_L^2) \tag{11b}$$

where





$G_s = G_{\mathrm{so}}(1 - e^{-a_s C_{\mathrm{ps}}})Z_s$ (12a)
$G_L = G_{\mathrm{L0}}(1 - e^{-a_l(C_{\mathrm{pl}}+Z_s)})Z_L$ (12b)
for small and large zooplankton respectively, $G_Z$ is grazing of small zooplankton by large
zooplankton, $R$ is respiration, and $m_1$ and $m_2$ are nongrazing mortality rates. Large zooplankton
grazing is divided into grazing on large phytoplankton and small zooplankton in proportion to
the relative abundance of each
$G_P = G_L \frac{P_l}{P_l+Z_s}$ (13a)
$G_Z = G_{lL} \frac{Z_s}{P_l+Z_s}$ (13b)
Zooplankton biomass loss to respiration is given by
$R = max\{r_z T_f Z - C_{\mathrm{XS}}. 0\}$ (14)
and uses the same activation energy as photosynthesis. Respiration ($R$) is assumed to consume
only carbon and not result in catabolism of existing biomass when "excess" carbon is available
in the prey. In addition, conservation of mass must be maintained by recycling to the dissolved
pool grazer consumption of elements in excess of biosynthetic requirements when grazer and
prey elemental ratios differ. In the case where the nutrient quota (relative to carbon) exceeds the
grazer fixed ratio, the excess nutrient is remineralized to the dissolved inorganic pool. In the case
where the nutrient quota is less than the grazer ratio, the grazer intake is reduced to what can be
supported by the least abundant nutrient (relative to the grazer biomass ratio) and excess carbon
is remineralized. For the case of two nutrients (in this case N and Fe) it is necessary to define
$G' = G\min\left\{\frac{N_P}{C_P} R_{\mathrm{CN}}, \frac{Fe_P}{C_P} R_{\mathrm{CFe}}, 1\right\}$ (15)





where $G$ is equal to $G_S$ (equation 12a) for small zooplankton and $G_P$ (equation 13a) for large
zooplankton, and $R_{XY}$ indicates the fixed ratio of element X to element Y in grazer biomass. The
'excess' carbon available for respiration is
$$C_{\text{XS}} = G'\left\{\frac{C_P}{N_P}R_{\text{NC}} - 1, \frac{C_P}{\text{Fe}_P}R_{\text{FeC}} - 1, 0\right\} \qquad (16a)$$
and the excess nutrients remineralized to their inorganic pools are
$$N_{\text{XS}} = G'\text{max}\left\{\frac{N_P}{C_P} - R_{\text{NC}}.0\right\}\varepsilon + G'\text{max}\left\{R_{\text{NC}}\left(\frac{N_P}{\text{Fe}_P}R_{\text{FeN}} - 1\right).0\right\}(1 - \varepsilon) \qquad (16b)$$
$$\text{Fe}_{\text{XS}} = G'\text{max}\left\{\frac{\text{Fe}_P}{C_P} - R_{\text{FeC}}.0\right\}\varepsilon + G'\text{max}\left\{R_{\text{FeC}}(\frac{\text{Fe}_P}{N_P}R_{\text{NFe}} - 1).0\right\}(1 - \varepsilon) \qquad (16c)$$
where
$$\varepsilon = \frac{\text{max}\{C_{\text{xs}}, 0\}}{C_{\text{xs}} + \Delta}$$
is a switch to prevent double-counting in cases where one of the terms is redundant (the excess
relative to the least abundant element is included in the other term), but would otherwise be
nonzero ($\Delta$ is a constant equal to $10^{-15}$, to prevent divide-by-zero). For three elements, there are
$3! = 6$ possible cases: for N greater or less than $C_PR_{\text{NC}}$, Fe may be either in excess relative to
both C and N, deficient relative to both, or in excess relative to one but not the other (Table 2).

Table 2 - Cases where the 'excess' terms are nonzero. These terms are always greater than or
equal to zero, and always zero when the phytoplankton elemental ratio is equal to the grazer bio-
mass ratio. A plus (+) sign indicates that a specific term is positive. $N_1$ and $N_2$, $Fe_1$ and $Fe_2$ indi-
cate the first and second terms in equations 16b and 16c. $R_{\text{NC}}$ is the grazer N/C (Redfield) ratio.

| | Fe in excess relative to both C and N | | | | Fe in excess relative to C or N but not both | | | | Fe deficient relative to both C and N | | | | |
|---|---|---|---|---|---|---|---|---|---|---|---|---|---|---|
| | C | $N_1$ | $N_2$ | $Fe_1$ | $Fe_2$ | C | $N_1$ | $N_2$ | $Fe_1$ | $Fe_2$ | C | $N_1$ | $N_2$ | $Fe_1$ | $Fe_2$ |
| N/C>$R_{\text{NC}}$ | | + | | + | | | + | | + | | + | | + | | |
| N/C<$R_{\text{NC}}$ | + | | | | + | + | | | | + | + | | + | | |

**2.3 Organic and Inorganic Pools**

There are two pools of detritus with different sinking rates but the same fixed elemental ratios.
Detrital C/N/Fe ratios are the same as zooplankton, so zooplankton mortality or grazing of small
zooplankton by large zooplankton produce no 'excess'. Phytoplankton mortality, and defecation
by zooplankton grazing on phytoplankton, produces excess nutrient or excess C that needs to be
recycled into the inorganic pool in a similar fashion as outlined above for the assimilated fraction
of grazing on phytoplankton.
The conservation equations for detrital C are
$$\frac{dD_s}{dt} = m_1(C_{ps} + Z_s) + m_2(C_{ps}^2 + Z_S^2) - r_1 D_s T_g - w_s \frac{dD_s}{dz} \qquad (17a)$$
$$\frac{dD_l}{dt} = m_1(C_{pl} + Z_L) + m_2(C_{pl}^2 + Z_L^2) - r_2 D_l T_g - w_l \frac{dD_l}{dz} \qquad (17b)$$
where $T_g$ is an Arrhenius function for temperature dependence of remineralization and $w$ is the
sinking speed. The conservation equations for inorganic C, N, and Fe are
$$\frac{dC_i}{dt} = (\zeta V^N - P_{phot}^C)C_p + R + C_{XS} + (r_1 D_s + r_2 D_l)T_g \qquad (18a)$$
$$\frac{dN_i}{dt} = -\frac{V^N}{Q^N} N_p \left(\frac{L_{NO3}}{L_{NO3}+L_{NH4}}\right) + N_{ox} - N_{dentr}(1 - A_f) \qquad (18b)$$
$$\frac{dN_a}{dt} = -\frac{V^N}{Q^N} N_p \left(\frac{L_{NH4}}{L_{NO3}+L_{NH4}}\right) + \frac{R}{R_{CN}} + N_{XS} + (r_1 D_s + r_2 D_l)R_{NC}T_g - N_{ox} + N_{dnf} - N_{dentr}A_f \qquad (18c)$$
$$\frac{dFe}{dt} = \frac{V^{Fe}}{Q^{Fe}} Fe_p + \frac{R}{R_{CFe}} + Fe_{XS} + (r_1 D_s + r_2 D_l)R_{FeC}T_g \qquad (18d)$$
where $N_{ox}$ is microbial oxidation of ammonium to nitrate (nitrification), $N_{dnf}$ and $N_{dentr}$ are
sources and sinks associated with dinitrogen fixation and denitrification, and $A_f$ is the ammonium
fraction of denitrification losses, associated with anaerobic ammonium oxidation ("anammox").





The oxygen equation is essentially the inverse of equation 18a, with additional terms for
oxidation and reduction of N, i.e.,
$\frac{dO_2}{dt} = -\frac{dC_i}{dt} + 2\frac{V^N}{Q^N}N_p(\frac{L_{NO3}}{L_{NO3}+L_{NH4}}) - 2N_{ox}$ (19)
Nitrification is given by
$N_{ox} = k_{NH4ox}N_a\frac{K_E}{K_E+E(z)}$ (20)
where $E(z)$ is the layer mean irradiance at depth $z$. Dinitrogen fixation is parameterized as an
external input of ammonium dependent on light, temperature and Fe availability, and inhibited
by high ambient concentrations of inorganic N,
$N_{dnf} = k_{dnf}T_{dnf}(1 - e^{-aE})(\frac{Fe}{K_{Fe}+Fe})(\frac{K_{NO3}}{K_{NO3}+N_i+N_a})$ (21)
where $T_{dnf}$=max(0, 1.962($T_f$ - 0.773)), i.e., a linear multiple of equation (1) that is 0 at T<20°C
and unity at T=30°C.

Denitrification is parameterized as a fraction of total remineralization that increases as a linear
function of oxygen concentration for concentrations less than a threshold concentration $O_{mxd}$
$N_{frxn} = 1 - \frac{min(O_2, O_{mxd})}{O_{mxd}}$ (22)
Remineralization is then divided among oxygen (1-$N_{frxn}$), nitrate (0.875$N_{frxn}$), and ammonium
(0.125$N_{frxn}$) assuming an average anammox contribution of 25% (Babbin et al., 2014). We use
this average ratio of anammox to classical denitrification to partition fixed N losses between
$NO_3^-$ and $NH_4^+$; the DIC sink and organic matter source associated with anammox are small and
are neglected here.



**2.4 Calcification, Calcite Dissolution, and Alkalinity**

Calcification is represented by a detrital calcium carbonate ($CaCO_3$) state variable, but no

explicit calcifier groups. Detrital $CaCO_3$ sinks in the same fashion as detrital particulate organic

carbon (POC), with a sinking rate independent of those for large and small organic detritus.

Calcite production is represented as a fixed fraction of detritus production from small

phytoplankton and small zooplankton mortality:

$$\frac{dCa}{dt} = m_1(C_{ps} + Z_s)P_{Ca} + m_2(C_{ps}^2 + Z_S^2)P_{Ca} - k_{Ca}Ca - w_{Ca}\frac{dCa}{dz} \tag{23}$$

Calcite dissolution occurs throughout the water column as a first order process (i.e., no

dependence on temperature or saturation state). Approximately 80% of calcite produced is

exported from the euphotic zone. Burial in the sediments is represented as a simple 'on/off'

switch dependent on the calcite saturation state (zero when $\Omega_C < 1$ and 1 when $\Omega_C \geq 1$). Calcite

burial is balanced by an equivalent source of DIC and alkalinity at the ocean surface as a crude

parameterization of fluvial sources.

For each mole of calcite production two moles of alkalinity equivalent are lost from the

dissolved phase; the reverse occurs during calcite dissolution. There are additional sources and

sinks for alkalinity associated with phytoplankton nutrient uptake, organic matter

remineralization, nitrification, denitrification and dinitrogen fixation (Wolf-Gladrow et al., 2007,

see Supplementary Table S2). The anammox reaction does not in itself contribute to alkalinity

(Jetten at al., 2001), but there is a sink associated with ammonium oxidation to nitrite (the model

does not distinguish between nitrite and nitrate). Autotrophic production of organic matter by

anammox bacteria is a net source of alkalinity (Strous et al., 1998) but this source is extremely



small (~0.03 mol/molN) and is neglected here. Globally, the sources and sinks of alkalinity from
the N cycle offset each other such that there is no net gain or loss as long as the global fixed N
pool is conserved (see below Sect. 2.5). If dinitrogen fixation and denitrification are allowed to
vary freely, there will generally be a net gain or loss of fixed N and, therefore, of alkalinity.

**2.5 External Nutrient Sources and Sinks**

External sources and sinks consist of river inputs, aeolian deposition, biological $N_2$ fixation, de-
nitrification, mobilization of Fe from reducing sediments, loss of Fe to scavenging, and burial of
calcium carbonate in the sediments. Aeolian deposition of Fe is calculated from a climatology of
mineral dust deposition generated from offline (atmosphere-only) simulations with CanAM4
(von Salzen et al., 2013), with an Fe mass fraction of 5% and a fractional solubility of 1.4% in
the surface layer. Subsurface dissolution is parameterized based on PISCES (Aumont et al.,
2015); the total dissolution is 6.35%, with 22% of soluble Fe input into the first vertical layer
(see Supplementary material). Iron from reducing sediments is also based on PISCES, with a
constant areal flux of 1000 nmol m$^{-2}$ d$^{-1}$ in the first model level, declining exponentially with in-
creasing seafloor depth (i.e., assuming that the sediments become progressively more oxygen-
ated) with an e-folding length scale of about 200 m. Scavenging of dissolved iron is first-order
with a high rate (2.5 d$^{-1}$) for concentrations in excess of 0.6 nM (Johnson et al., 1997). For con-
centrations below this threshold, the rate is much lower (0.001 d$^{-1}$) and is weighted by the con-
centration of organic detritus (Christian et al., 2002b), i.e.,
$\frac{dFe}{dt} = -FeS_{Fe1}min\{(D_S + D_L)P_{Fe}, 1\}$                                                         (24)



where Fe is the dissolved iron concentration, $D_S$ and $D_L$ are the small and large detritus concen-
trations, $S_{Fe1}$ is the first-order scavenging rate in surface waters with abundant particulates, and
$P_{Fe}$ is an empirical parameter to determine the dependence on particle concentration (Table 1).
The basis for this parameterization is that the rate of scavenging must depend not only on the
concentration of iron but on the concentration of particles available for it to precipitate onto, and
assumes that POC is strongly positively correlated with total particulate matter. Scavenging is
treated as irreversible, i.e., scavenged Fe is not tracked and does not reenter the dissolved phase.

Unlike in CMOC, $N_2$ fixation and denitrification vary independently in CanOE, so the global
total N pool can change. Conservation is imposed by adjusting the global total N pool according
to the difference between the gain from $N_2$ fixation and the loss to denitrification. A slight
adjustment is applied to the nitrate concentration at every grid point, while preserving the overall
spatial structure of the nitrate field. Adjustments are multiplicative rather than additive to avoid
producing negative concentrations. This adjustment does not maintain (to machine precision) a
constant global N inventory but is intended to minimize long term drift, keeping it much smaller
than the free surface error (see below). This adjustment is applied every 10 days and has a
magnitude of approximately $7 \times 10^{-8}$ of the total N.

One mole of alkalinity is removed per mole of N added or removed, since there are alkalinity
sources of 1 mol/molN associated with both $N_2$ fixation (creation of new $NH_4^+$) and
denitrification (removal of $NO_3^-$), offset by a 2 mol/molN sink associated with nitrification. As
noted above, $CaCO_3$ can dissolve or be buried in the sediments depending on the calcite
saturation. DIC and alkalinity lost to burial are reintroduced at the ocean surface, at the same grid





point as burial occurs, providing a crude parameterization of river inputs so that global
conservation is maintained (fresh water runoff contains no DIC or alkalinity). However, the OPA
free surface formulation is inherently imperfect with regard to tracer conservation. Drift in total
ocean alkalinity and nitrogen over time is on the order of 0.01% and 0.03% per thousand years,
respectively (losses due to the free surface are generally larger for tracers with less homogeneous
distributions).

**2.6 Ancillary data**

For first-order model validation we have relied largely on global gridded data products rather
than individual profile data. Global gridded data from World Ocean Atlas 2013 (WOA2013)
(Locarnini et al., 2013; Zweng et al., 2013; Garcia et al., 2014a; 2014b) were used for
temperature, salinity, and oxygen and nitrate concentration. DIC and alkalinity were taken from
the GLODAP gridded data product (Lauvset et al., 2016). Offline carbon chemistry calculations
were done following the Best Practices Guide (Dickson et al., 2007) and the OMIP-BGC
protocols (Orr et al., 2017), which are identical to those used in the models except that constant
reference concentrations were used for phosphate (1 µM) and silicate (10 µM).

There is no global gridded data product for Fe, but we have made use of the GEOTRACES
Intermediate Data Product 2017 (Schlitzer et al., 2018), and the data compilations from MBARI
(Johnson et al., 1997; 2003) and PICES Working Group 22 (Takeda et al., 2013). The latter two
are concentrated in the Pacific, while GEOTRACES is more global. The combined data sets
provide more than 10000 bottle samples from more than 1000 different locations (Supplementary





Figure S4a) (excluding some surface transect data that involve frequent sampling of closely
spaced locations along the ship track). More detail about model comparison to these data
compilations and the list of original references are given in the Supplementary information.

Satellite ocean colour estimates of surface chlorophyll were taken from the combined
SeaWiFS/MODIS climatology described by Tesdal et al. (2016). Climatological satellite POC
was downloaded from the NASA ocean colour web site and is based on the algorithm of
Stramski et al. (2008) using MODIS-Aqua data. This climatology differs slightly from the
chlorophyll one in terms of years included and sensors utilized, but as only climatological
concentrations are considered and each climatology covers ~15 years, these differences will have
negligible effect on the results presented.

CMIP6 model data were regridded to a common grid (2x2°, 33 levels following the GLODAP
levels) to facilitate ensemble averaging. The years 1986-2005 of the Historical experiment were
averaged into climatologies or annual means, for meaningful comparison with observation-based
data products. A single realization was used in each case; as 20 year averages are used, internal
variability is assumed to have little effect (e.g., Arguez and Vose, 2011). Sampling among
CMIP6 models was somewhat opportunistic and the exact suite of models varies among the
analyses presented. When we conducted a search for a particular data field, we included in the
search parameters all models that published that field, and repeated the search at least once for
models that were unavailable the first time the search was executed. In some cases, model
ensemble means excluded all but one model from a particular 'family' (e.g., there are three
different MPI-ESM models for which ocean biogeochemistry fields were published), as the



solutions were found to be similar and would bias the ensemble mean towards their particular
climate. More detail is given in Supplementary Table S3.

**3. Results**

**3.1 Distribution of oxygen**

The spatial distribution of oxygen concentration ($[O_2]$) at selected intermediate depths (400, 900,
and 1400 m) for CanESM5, CanESM5-CanOE, a model ensemble mean (MEM) of CMIP6
models (excluding CanESM5 and CanESM5-CanOE), and gridded data from WOA2013 is
shown in Figure 2. The major features are consistent across the models. All three cases show
elevated oxygen concentrations relative to observations, particularly in the North Pacific, the
North Atlantic and the Southern Ocean. In the Indian Ocean, both CanESM models show high
oxygen concentrations in the Arabian Sea and deeper layers of the Bay of Bengal relative to
observations and other CMIP6 models; these biases are somewhat smaller in CanESM5-CanOE
than in CanESM5 (Figure 2).

Biases in the eastern boundary current regions are depth and model specific. CanESM5 shows
particularly strong oxygen depletion at 1400 m in the eastern tropical Pacific. In the southeastern
Atlantic, models tend to be biased low at the shallower depths, and show somewhat more
variation at greater depths (Figure 2). Overall, $[O_2]$ biases tend to be positive over large areas of
ocean with the exception of some eastern boundary current regions, implying that models
exaggerate the extent to which remineralization is concentrated in these regions. An alternate





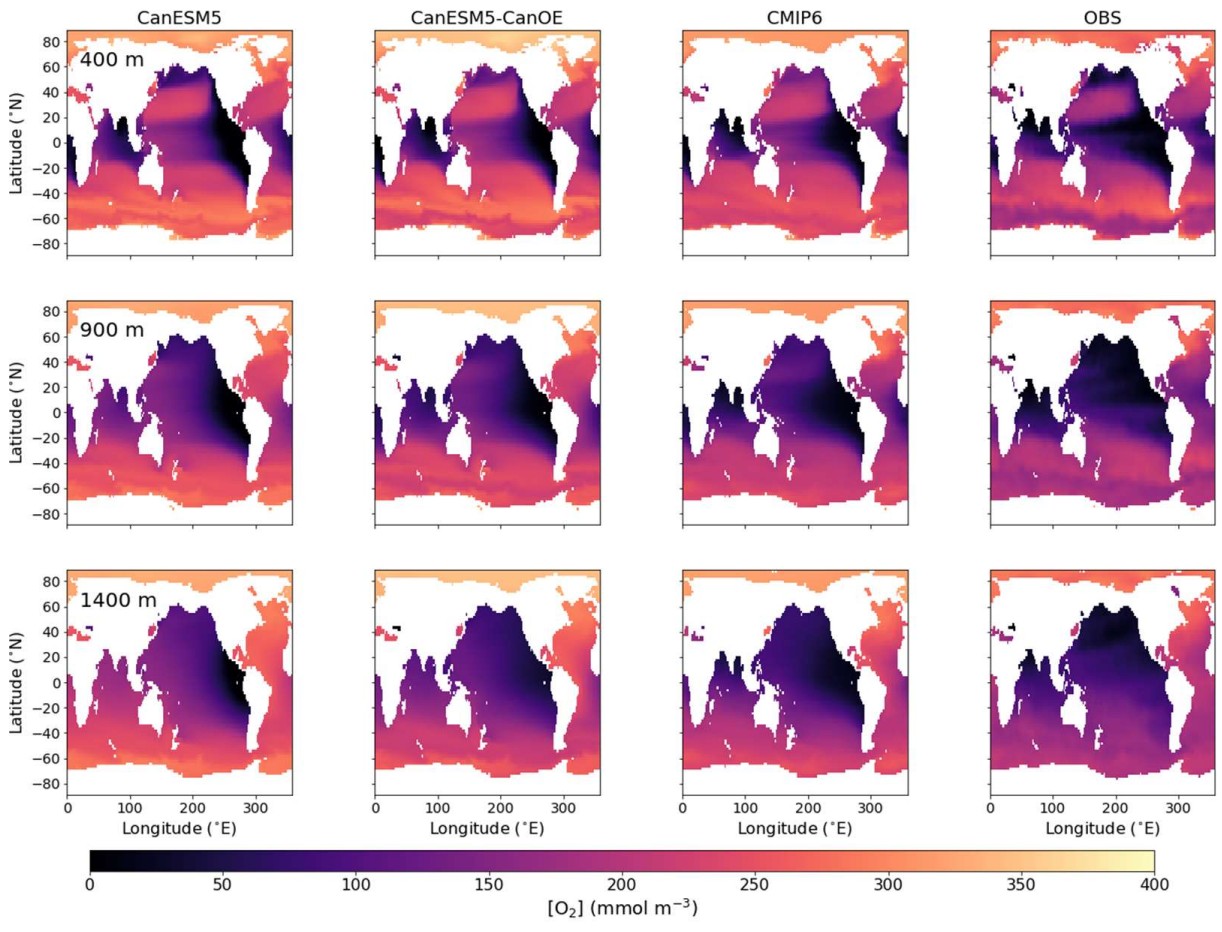

Figure 2 - Global distribution of oxygen ($O_2$) concentration in mmol $m^{-3}$ at 400, 900, and 1400 m (rows) for CanESM5-CanOE, CanESM5, the mean for other (non-CanESM) CMIP6 models, and World Ocean Atlas 2013 (WOA2013) observations (columns). Difference from the observation-based fields are shown in Supplementary Figure S3.



version of Figure 2 that shows model errors relative to the observational data product is given in
Supplementary Figure S3.

The zonal mean oxygen concentration, saturation concentration, and apparent oxygen utilization
(AOU) are shown in Figure 3 for the same four cases. Again, the models generally show a
positive bias in $[O_2]$, particularly in high-latitude deep waters. The major ocean circulation
features are reproduced fairly well in all cases (e.g., weaker ventilation of low-latitude
subsurface waters, greater vertical extent of well-ventilated surface waters in the subtropics). The
saturation concentration (a function of temperature and salinity) generally shows relatively little
bias, implying that the bias in $[O_2]$ arises mainly from remineralization and/or ventilation. AOU
is lower than observed over much of the subsurface ocean. Regional biases are quite consistent
across models, but are slightly greater in CanESM5 than in CanESM5-CanOE or the ensemble
mean, except in the Arctic Ocean. Again, Supplementary Figure S3 includes a version of this
plot that shows the model differences from the observations.

The skill of each model with respect to the distribution of $O_2$ at different depths is represented by
Taylor diagrams (Taylor, 2001) in Figure 4, in which all of the CMIP6 models that were shown
as an ensemble mean in Figures 2 and 3 are shown individually. The blue dots represent
CanESM5, red CanESM5-CanOE, and grey the ensemble mean of all CMIP6 models except
CanESM5 and CanESM5-CanOE; the smaller grey dots represent the individual models.
CanESM5-CanOE shows slightly higher pattern correlation than CanESM5 at all depths. Both
models compare favourably with the full suite of CMIP6 models, with r>0.85 for CanESM5 and
r>0.9 for CanESM5-CanOE at all depths examined, and a normalized standard deviation within



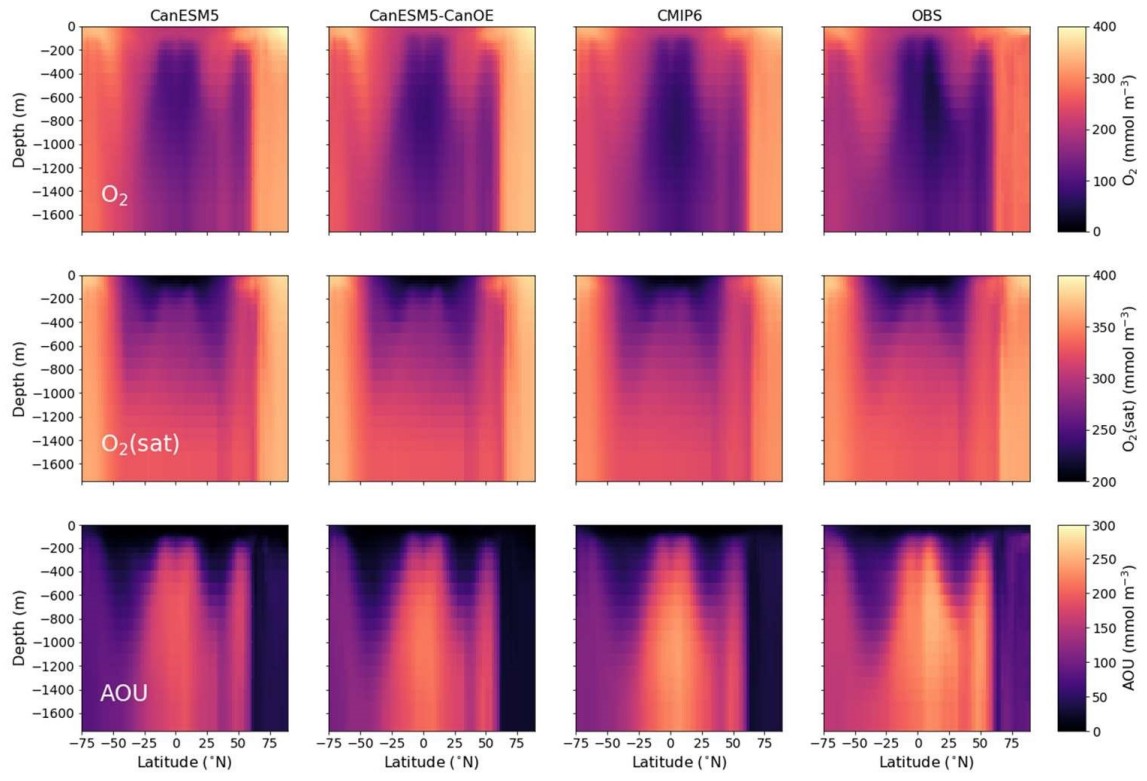

Figure 3 - Latitude-depth distribution (surface to 1750 m) of zonal mean oxygen concentration ($O_2$), oxygen concentration at saturation ($O_2$(sat)), and apparent oxygen utilization (AOU) in mmol m$^{-3}$ for CanESM5-CanOE, CanESM5, the mean for other CMIP6 models, and observations (WOA2013). Note different colour scales for different rows. Difference from the observation-based fields are shown in Supplementary Figure S3.





Figure 4 - Taylor diagrams (Taylor, 2001) comparing modelled and observed distributions of oxygen at specific depths from 100 to 3500 m. Angle from the vertical indicates spatial pattern correlation. Distance from the origin indicates ratio of standard deviation in modelled vs. observed (WOA2013) fields. Red dots represent CanESM5-CanOE, blue dots CanESM5, small grey dots other CMIP6 models, and large grey dots the model ensemble mean for all CMIP6 models except CanESM5 and CanESM5-CanOE.





±25% of unity.

The total volume of ocean with $[O_2]$ less than 6 mmol m$^{-3}$ (the threshold for denitrification
(Devol, 2008)) and 60 mmol m$^{-3}$ (a commonly used index of hypoxia) is shown in Figure 5. The
total volume is highly variable among models (note, however, that there are several clusters of
related models with quite similar totals). CanESM5 and CanESM5-CanOE have among the
lowest total volumes (i.e., the interior ocean is relatively well ventilated) and are among the
nearest to the observed total. For $[O_2]$ <60 mmol m$^{-3}$ the bias is, nonetheless, quite large (i.e., the
observed volume is underestimated by almost 50% in both models). The volume of water with
$[O_2]$ below the denitrification threshold is overestimated in both CanESM5 and CanESM5-
CanOE; CanESM5-CanOE has a much smaller total that is closer to the observed value. The bias
in the spatial pattern of hypoxia (not shown) is generally similar to the bias in dissolved oxygen
distribution (Figure 2). The low-oxygen regions are generally more concentrated in the eastern
tropical Pacific in the models than in observations, and the low-oxygen region in the northwest
Pacific is not well reproduced in CanESM models.

**3.2 Distribution of DIC, alkalinity, and CaCO$_3$ saturation**

The spatial distribution of aragonite saturation state ($\Omega_A$) at selected depths is shown in Figure 6
(the first two depths are the same as in Figure 2, but the third is much deeper). In this case the
observations are a combination of GLODAP (Lauvset et al., 2016) for DIC and alkalinity, and
WOA2013 for temperature and salinity. CanESM5 and CanESM5-CanOE generally compare
well with other models and observations. The low saturation bias in the eastern tropical Pacific is



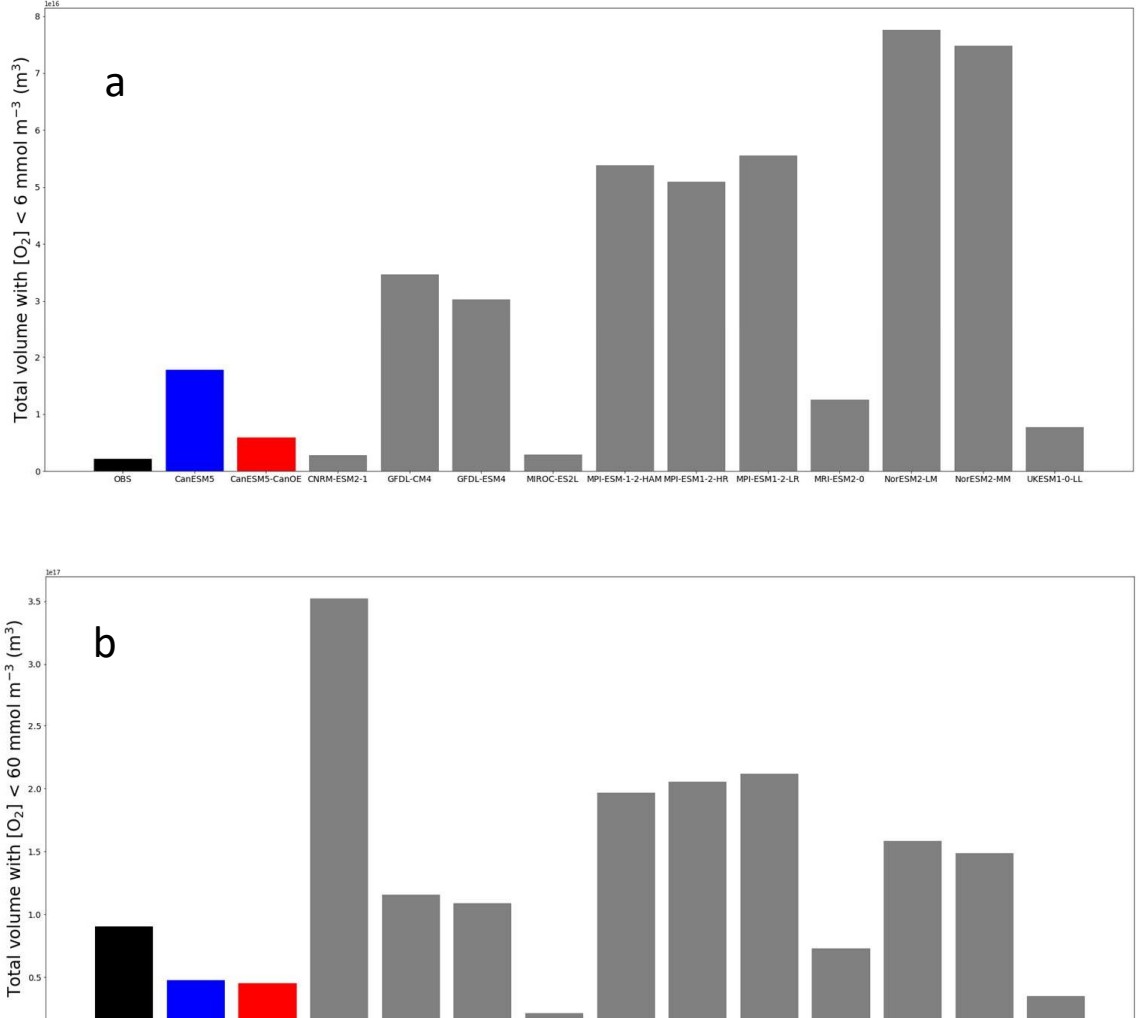

Figure 5 - Total volume of ocean with oxygen ($O_2$) concentration less than (a) 6 mmol $m^{-3}$ (mean for last 30 years of the historical experiment) and (b) 60 mmol $m^{-3}$. Observation are from WOA2013.



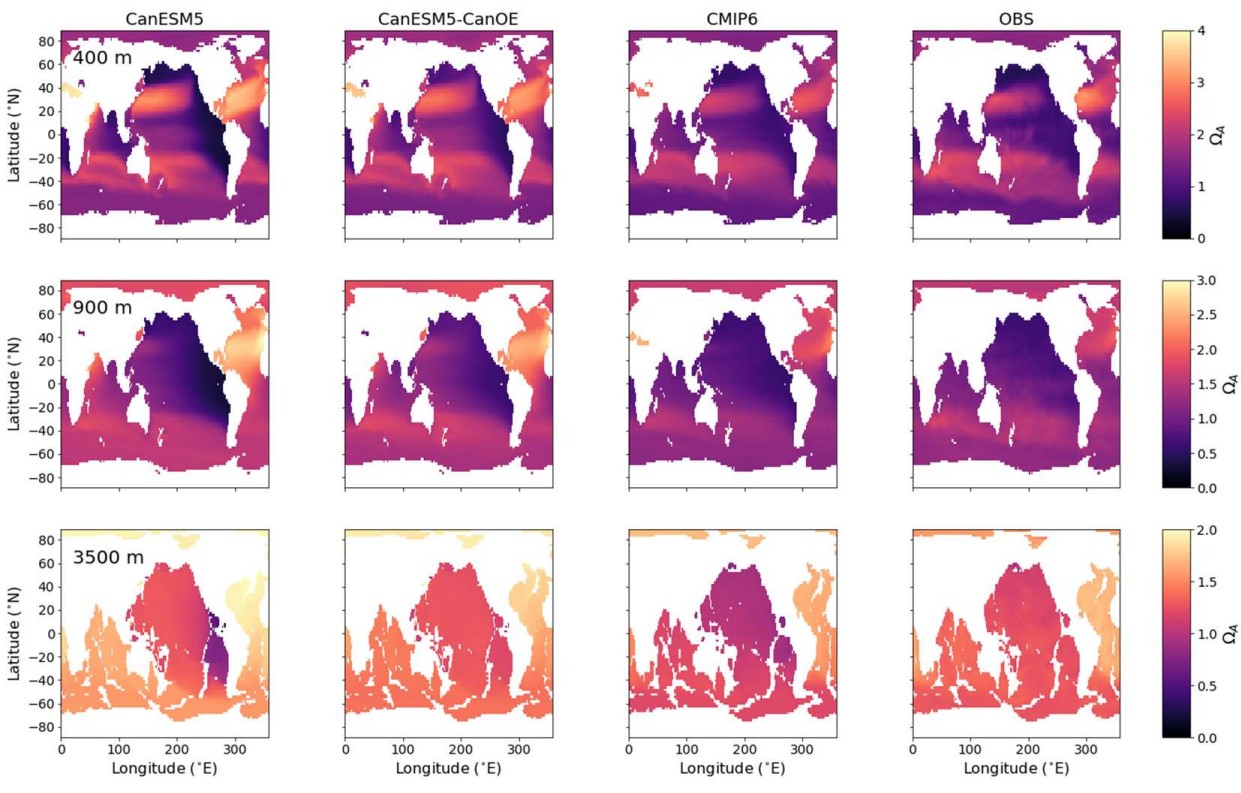

Figure 6 - Global distribution of aragonite saturation ($\Omega_A$) at 400, 900, and 3500 m for CanESM5-CanOE, CanESM5, the mean for other CMIP6 models, and observations (GLODAP + WOA2013). Note different colour scales for different depths. Difference from the observation-based fields are shown in Supplementary Figure S3.



substantially reduced in CanESM5-CanOE compared to CanESM5. On the other hand CanESM5
generally does better than CanESM5-CanOE, or the ensemble mean, at reproducing the low
saturation states in the northwestern Pacific and the Bering Sea. Both models show a high
saturation state bias in the North Atlantic and the well-ventilated regions of the north Pacific
subtropical gyre. These biases are reduced slightly in CanESM5-CanOE, probably due to the
smaller average remineralization length scale for organic detritus.

Zonal mean distributions of aragonite saturation state ($\Omega_A$), calcite saturation state ($\Omega_C$), and
carbonate ion concentration ($[CO_3^{--}]$) are shown in Figure 7 (Supplementary Figure S3 includes
versions of Figures 6 and 7 that explicitly show the model differences from the observations).
The models generally compare well with the observations in the representation of the
latitude/depth distribution of high and low saturation waters. CanESM5 has a high saturation bias
in low-latitude surface waters that is somewhat reduced in CanESM5-CanOE.  Both CanESM5
models show a high saturation bias in Northern Hemisphere intermediate (e.g., 200-1000 m)
depth waters that is larger than in the ensemble mean.

Taylor diagrams for a range of depths are shown for DIC in Figure 8 and for $\Omega_A$ in Figure 9 (for
alkalinity, see Supplementary Figure S2). As expected, the MEM generally compares favourably
with the individual models (e.g., Lambert and Boer, 2001). CanESM5 and CanESM5-CanOE
compare favourably with the full suite of CMIP6 models. CanESM5-CanOE shows a gain in
skill relative to CanESM5, and both show improvement relative to CanESM2. At 400 m,
CanESM2 stands out as having extremely high variance, which is mostly due to extremely high
DIC concentrations occurring over a limited area in the eastern equatorial Pacific (not shown).





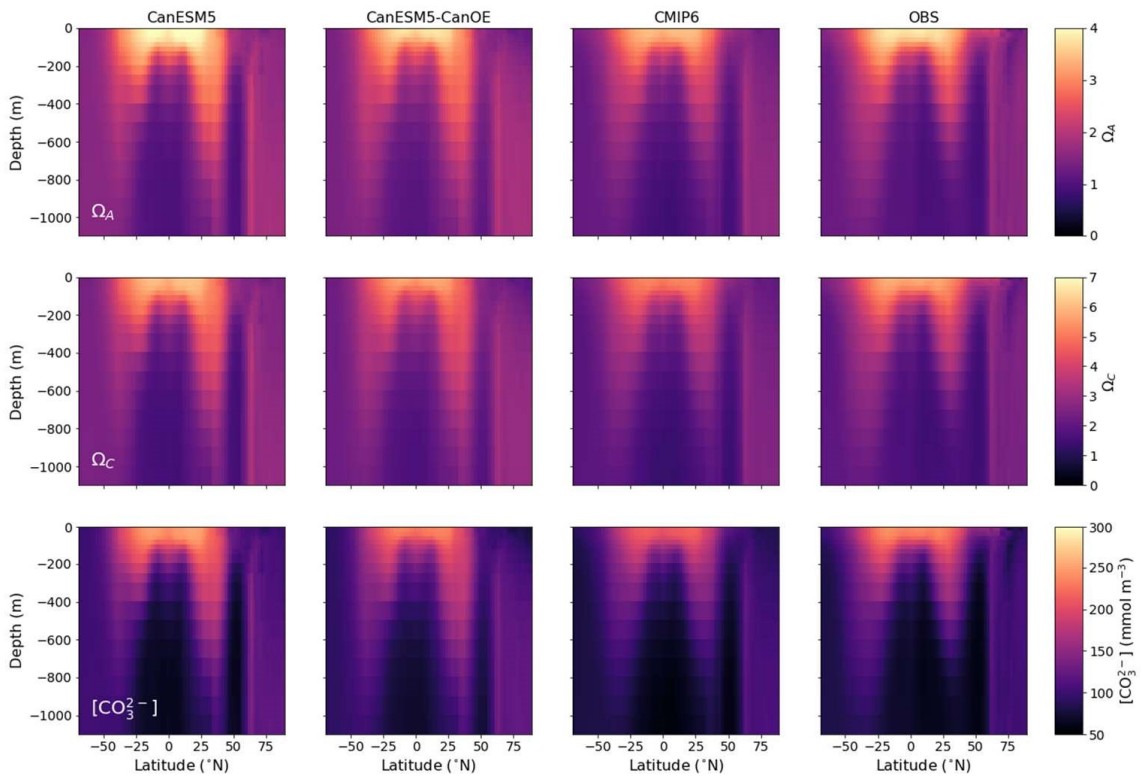

Figure 7 - Latitude-depth distribution of zonal mean (surface to 1150 m) aragonite saturation state ($\Omega_A$), calcite saturation state ($\Omega_C$), and carbonate ion concentration ($[CO_3^{--}]$) in mmol m$^{-3}$ for CanESM5-CanOE, CanESM5, the mean for other CMIP6 models, and observations (GLODAP + WOA2013). Difference from the observation-based fields are shown in Supplementary Figure S3.





Figure 8 - Taylor diagrams comparing modelled and observed distributions of DIC at specific depths from 100 to 3500 m. Observations are from GLODAP (Lauvset et al., 2016). Red dots represent CanESM5-CanOE, blue dots CanESM5, magenta dots CanESM2, small grey dots other CMIP6 models, and large grey dots the model ensemble mean for all CMIP6 models except CanESM5 and CanESM5-CanOE.





Figure 9 - Taylor diagrams comparing modelled and observed (GLODAP + WOA2013) distributions of $\Omega_A$ at specific depths from 100 to 3500 m. Symbol colours as in Figure 8.



This bias is present in CanESM5 and in CMIP6 models generally (Figure 6) but involves much
lower concentrations spread over a larger area.

**3.3 N and Fe cycles**

An important difference between CMOC and CanOE is the inclusion of a prognostic Fe cycle.
The CMOC iron mask (Zahariev et al., 2008) was a pragmatic solution in the face of resource
limitations but is inherently compromised as it can not evolve with a changing climate. Other
centres that introduced a prognostic Fe cycle between CMIP5 and CMIP6 include JAMSTEC
(MIROC-ESM, MIROC-ES2L) and the UK Met Office (HadGEM2-ES, UKESM1-0-LL). The
first order test of a model with prognostic, interacting Fe and N cycles is whether it can
reproduce the distribution of High-Nutrient, Low-Chlorophyll (HNLC) regions and the
approximate surface macronutrient concentrations within these. CanESM5-CanOE succeeded by
this standard, although the surface nitrate concentrations are biased low in the subarctic Pacific
and equatorial Pacific and high in the Southern Ocean and in the global mean (Figure 10).

The seasonal cycle of the zonal mean surface nitrate concentration for a selection of CMIP6
models is shown in Figure 11. CanESM5, CanESM5-CanOE, and CNRM-ESM2-1 reproduce the
equatorial enrichment and the low concentrations in the tropical-subtropical latitudes fairly well.
Some models either have very weak equatorial enrichment (MPI-ESM1-2-LR) or too high a
concentration in the off-equatorial regions (UKESM1-0-LL, NorESM2-LM). UKESM1-0-LL
has very high concentrations throughout the low-latitude Pacific, which biases the ensemble
mean (Figure 11a). Figure 11b shows the same data as Figure 11a but for a more limited latitude





Figure 10 - Climatological seasonal cycle of surface nitrate concentration averaged for selected ocean regions. Thick red line represents CanESM5-CanOE, thick blue line CanESM5, thick black line observations (WOA2013), thin grey lines individual CMIP6 models, and thick grey line the model ensemble mean (excluding CanESM5 and CanESM5-CanOE). Regional boundaries are given in Supplementary Table S4.







Figure 11 - (a) Climatological seasonal cycle of zonal mean surface nitrate concentration, for a selection of CMIP6 models, a model ensemble mean (MEM) excluding CanESM5 and CanESM5-CanOE, and an observation-based data product (WOA2013).



Figure 11 (b) As Figure 11(a) but for latitudes <20°.



range to better illustrate model behaviour in the tropics. CanESM5, CanESM5-CanOE, and
CNRM-ESM2-1 reproduce the seasonal cycle of tropical upwelling (e.g., Philander and Chao,
1991), with highest concentrations in summer.

The surface distribution of dissolved iron (dFe) in various CMIP6 models is shown in Figure 12.
For Fe there is no observation-based global climatology with which to compare the model
solutions (some comparisons to available profile data are shown in Supplementary Figures S4b-
g). CanESM5-CanOE shows a similar overall spatial pattern to other models, and generally falls
in the middle of the spread, particularly regarding concentrations in the Southern Ocean. Several
models show extremely high concentrations in the tropical-subtropical North Atlantic (Sahara
outflow region). CanESM5-CanOE, along with CNRM-ESM2-1 and CESM2, has much less
elevated concentrations in this region, due to lower deposition or greater scavenging or both.
CanESM5-CanOE has its lowest concentration in the eastern subtropical South Pacific, which is
common to many models (Figure 12). The area of strong surface depletion is generally more
spatially restricted in CanESM5-CanOE than in other models, and surface dFe concentrations are
greater over large areas of the Pacific. Both the north-south and east-west asymmetry of
distribution in the Pacific is greater in CanESM5-CanOE than in most other models, some of
which show the South Pacific minimum extending westward across the entire basin, and others
into the Northern Hemisphere. Only in CESM2 is this minimum similarly limited to the
southeast Pacific.

The mean depth profiles of dFe are shown in Figure 13. Some models show more of a "nutrient-
type" (increasing with depth due to strong near-surface biological uptake and subsequent





Figure 12 - Global distribution of dissolved iron (dFe) concentration (log10 of concentration in nmol m$^{-3}$) at the ocean surface for CanESM5-CanOE and other CMIP6 models that published this field. Concentrations exceeding 1000 nmol m$^{-3}$ are masked white.



Figure 13 - Global mean depth profiles of dissolved iron concentration for CanESM5-CanOE and other CMIP6 models that published this field. GFDL-CM4 is excluded because it has very high concentrations (>2000 nmol m$^{-3}$) near the surface. Thick red line represents CanESM5-CanOE, thin grey lines individual CMIP6 models, and the thick grey line the model ensemble mean (excluding CanESM5-CanOE and GFDL-CM4).



remineralization) profile, some a more "scavenged-type" (maximal at the surface, declining with
depth) profile (cf. Li, 1991; Nozaki, 2001), and others a hybrid profile (increasing downward but
with a surface enrichment). CanESM5-CanOE is at the "nutrient-type" end of spectrum with a
generally monotonic increase with depth to a near-constant deep-water concentration of 0.6 nM
and a very slight near-surface enrichment (see also Supplementary Figures S4b,c). In CanESM5-
CanOE the scavenging model is very simple, with distinct regimes for concentrations greater or
less than 0.6 nM; scavenging rates are very high above this threshold which causes deep-water
concentrations to converge on this value. The generally nutrient-like profile suggest that in
CanOE the scavenging rate is quite low for concentrations below 0.6 nM. CanOE considers
particulate organic matter (POM) as an index of all particulate matter available for scavenging
onto, and model POM concentrations fall off rapidly below the euphotic zone (not shown).

Mean surface nitrate and dFe concentrations for selected ocean regions are shown in Figure 14.
CanESM5-CanOE shows concentrations that are within the range of CMIP6 models, although in
some cases at the higher or lower end. Surface nitrate concentrations generally compare
favourably with the observation-based climatology, but are biased low in HNLC regions other
than the Southern Ocean. These biases are not necessarily a consequence of having too much or
too little iron. For example, in the Southern Ocean CanESM5-CanOE has among the highest
surface nitrate concentrations, but it also has some of the highest dFe concentrations, and the
high nitrate bias is present in CanESM5 as well. Comparisons with the limited GEOTRACES
data available suggest that near surface dFe concentrations in the Southern Ocean are biased high
rather than low in CanESM5-CanOE (not shown). One region where there does seem to be a
strong correlation between surface nitrate and dFe concentrations is the western subarctic





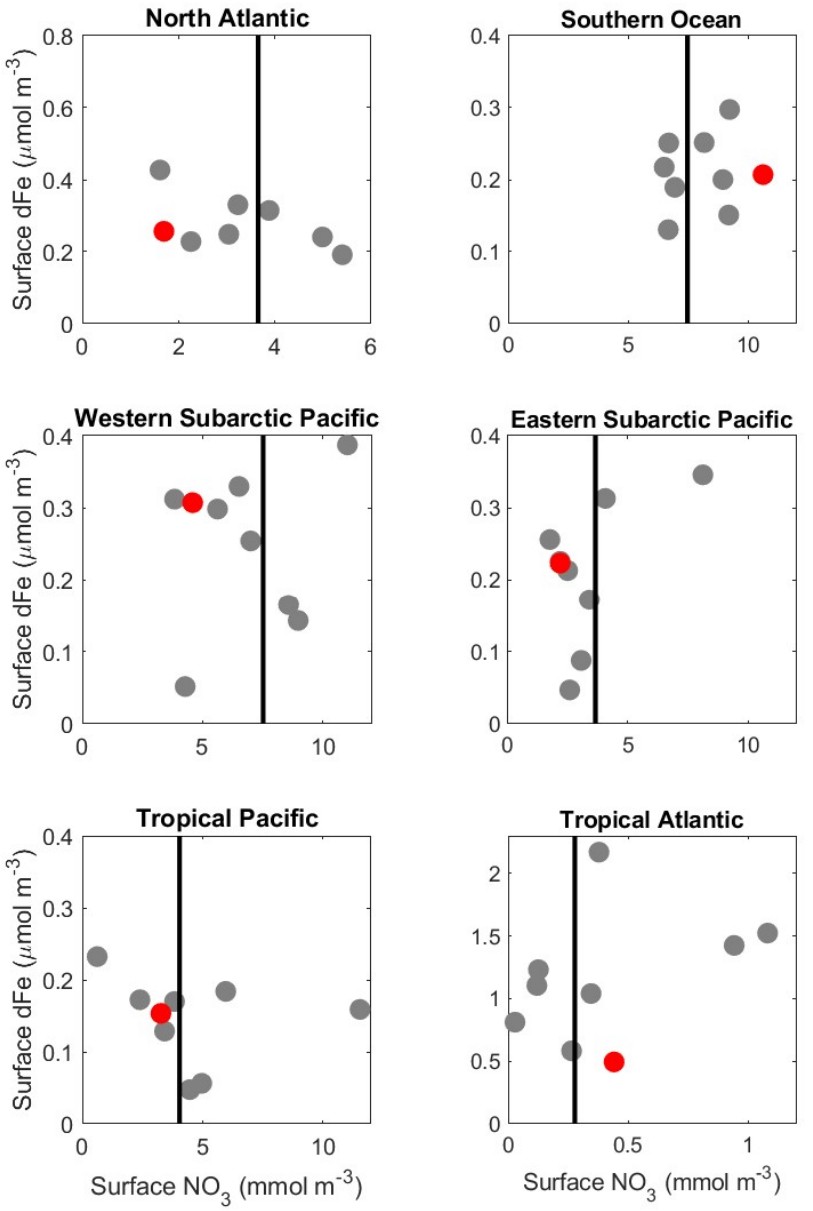

Figure 14 - Mean surface nitrate (NO₃) vs. dissolved iron (dFe) concentrations in different oceans, including the major high nutrient / low chlorophyll (HNLC) regions. CanESM5-CanOE is shown as a red dot and other CMIP5 models as grey dots (CanESM5 is not included because it does not have iron). Observed NO₃ is shown as a vertical black line as there are no observational estimates of dFe concentration. Region definitions are given in Supplementary Table S4.



Pacific. All but two models fall along a spectrum from high Fe / low nitrate to low Fe / high
nitrate. CanESM5-CanOE falls near the high Fe / low nitrate end of the range.

Surface nitrate concentrations along the Pacific equator during the upwelling season (June-
October) for CanESM5 and CanESM5-CanOE are shown in Figure 15. CanESM5-CanOE better
represents the east-west gradient, while CanESM5 has slightly higher concentrations in the core
upwelling region. Both models underestimate the highest concentrations around 100°W. Some
localized maxima in this data product are due to undersampling; however, examination of
ancillary data sets such as satellite sea surface temperature suggests that the enrichment at
100°W accurately reflects ocean upwelling (not shown). Although CanESM5 iron limitation is
calculated from an earlier version of the same data product, the Fe mask is based on the
minimum nitrate concentration over the annual cycle, whereas the data shown here are for the
upwelling season. In CanESM5-CanOE, the distribution of surface nitrate is an emergent
property of the model, and the fidelity to the observed distribution is generally good.

**3.4 Plankton biomass, detritus, and particle flux**

The relative abundance of the four living plankton groups are shown in Figure 16 for a range of
ocean regions. Both CanESM models mostly compare favourably with observation-based
estimates of phytoplankton biomass, except in the tropics where CanESM5-CanOE has very high
biomass. Both CanESM models have low phytoplankton biomass in the North Atlantic. In the
North Pacific and the Southern Ocean, CanESM5-CanOE reproduces the observation-based
estimates well, and CanESM5 slightly less well. The general pattern is that large and small





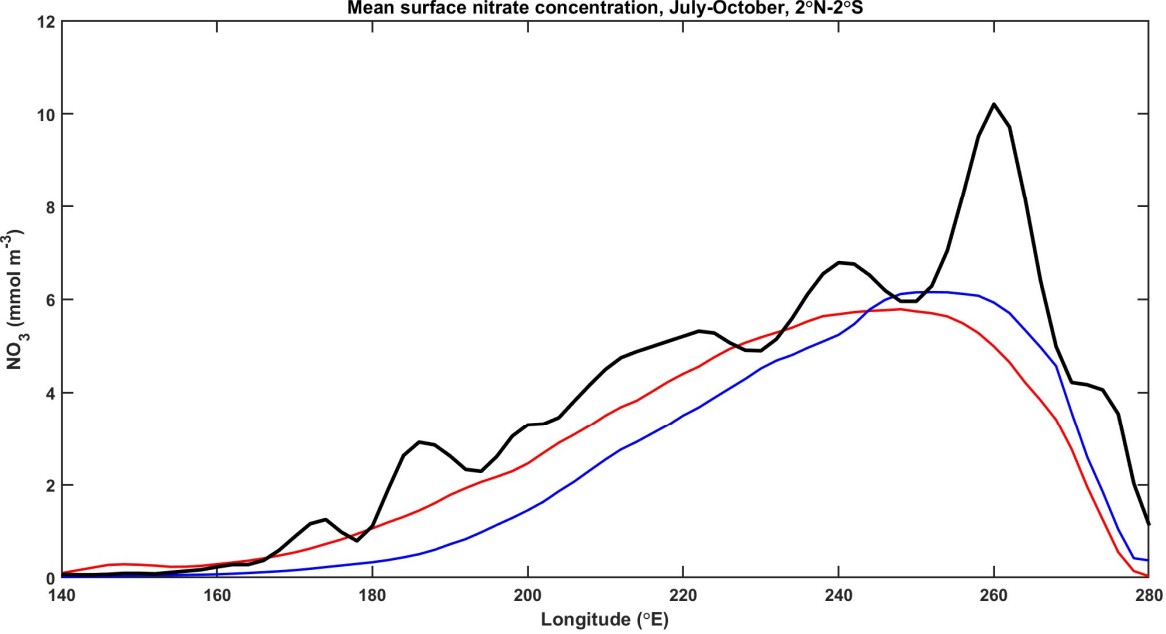

Figure 15 - Surface nitrate (NO3) concentrations along the Pacific equator (mean from 2°S-2°N) during the upwelling season (June-October) for CanESM5-CanOE (red), CanESM5 (blue), and WOA2013 observations (black).



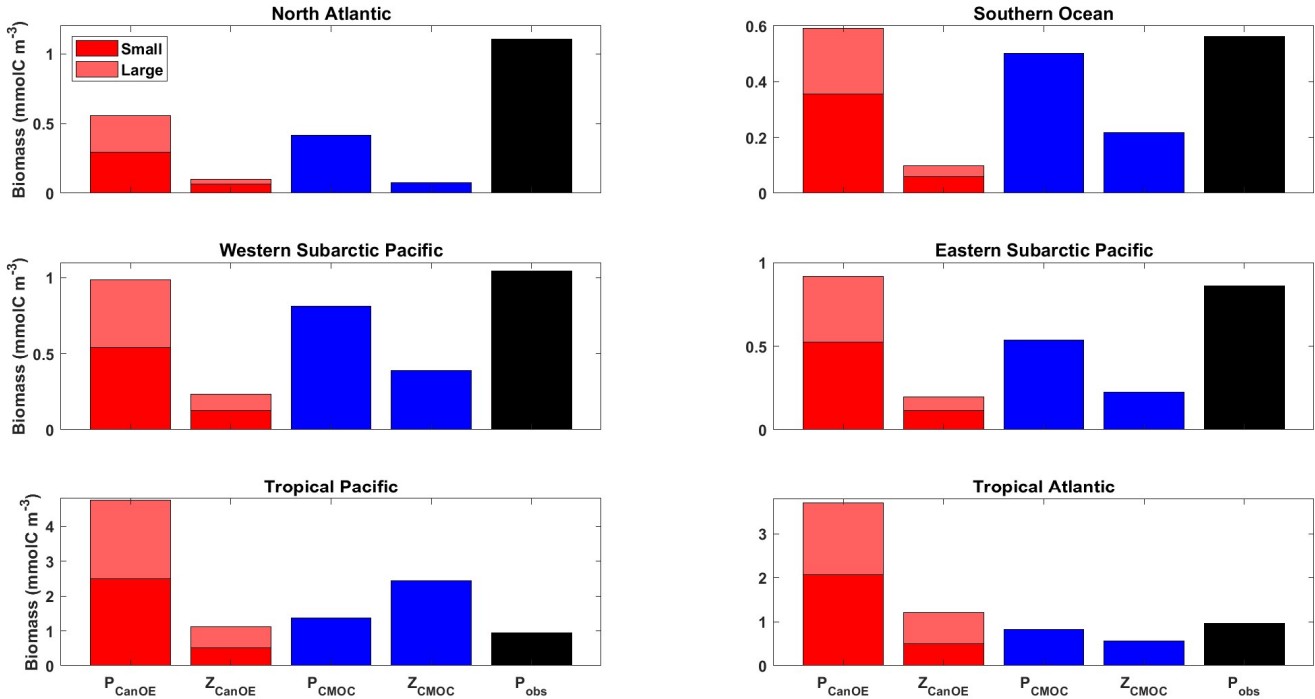

Figure 16 - Annual mean surface ocean concentration of large and small phytoplankton and zooplankton in CanESM5-CanOE (red) and of phytoplankton and zooplankton in CanESM5 (blue) for the representative ocean regions shown in Figure 14. Observational estimates (black) are for phytoplankton biomass calculated from satellite ocean colour estimates of surface chlorophyll (SeaWiFS/MODIS; Tesdal et al. 2016), assuming a carbon-to-chlorophyll ratio of 50 g/g. Region definitions are given in Supplementary Table S4.





phytoplankton have similar abundance, and are substantially more abundant than zooplankton.
Part of the rationale for multiple food chains is that they better represent the way that actual
plankton communities adapt to different physical ocean regimes and therefore are better able to
simulate distinct ocean regions with a single parameter set (e.g., Chisholm, 1992; Armstrong,
1994; Landry et al., 1997; Friedrichs et al., 2007). The expectation is that small phytoplankton
will be more temporally stable and large phytoplankton will fluctuate more strongly between
high and low abundances.

The mean annual cycles of surface chlorophyll largely conform to the expected pattern, e.g., in
the North Atlantic and the western subarctic Pacific large phytoplankton are dominant in summer
and much more variable over the seasons (Figure 17). Compared to observations, CanESM5
models underestimate the amplitude of the seasonal cycle in the North Atlantic and overestimate
it in the North Pacific. CanESM5 shows a stronger and earlier North Atlantic spring bloom
compared to CanESM5-CanOE; the observations are in between the two in terms of timing, and
both models underestimate the amplitude (Figure 17). In the tropics, the seasonal cycle is weak.
The tropical Atlantic shows the expected seasonal cycle but not the expected dominance of large
phytoplankton in summer. These size-fractionation patterns are difficult to validate against
observations. CanESM5-CanOE generally overestimates the total near surface chlorophyll in
both the tropical Pacific and the tropical Atlantic.

Zooplankton biomass (especially microzooplankton) is also somewhat difficult to test against
observations, but our model concentrations appear to be biased low. White et al. (1995), for
example, show a transect of vertically resolved mesozooplankton abundance along 140°W in the




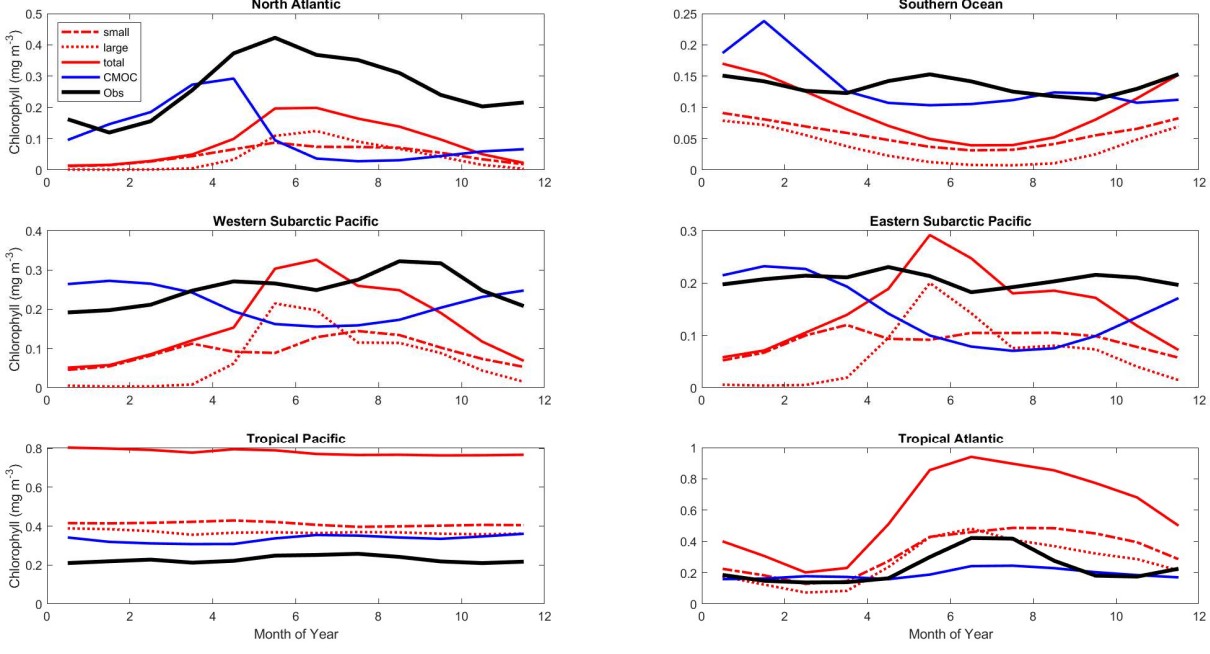

Figure 17 - Mean annual cycle of surface chlorophyll for the representative ocean regions shown in Figures 14 and 16. CanESM5-CanOE large and small phytoplankton concentrations are shown separately and combined (red) along with CanESM5 (blue) and observational estimates (black). Region definitions are shown in Supplementary Table S4.





tropical Pacific, with biomass ranging from about 0.1-0.7 mmolC m$^3$; CanESM5-CanOE
concentrations in this region are much lower. Stock et al. (2014) estimated depth-integrated
biomass of phytoplankton, mesozooplankton, and microzooplankton for a range of oceanic
locations in which intensive field campaigns have occurred (estimates of microzooplankton
biomass are relatively sparse). They found that in most locations phytoplankton and (combined)
zooplankton biomass are of comparable magnitude, whereas in CanESM5-CanOE zooplankton
biomass is consistently lower (Figure 16). The global integral biomass of mesozooplankton is
about an order of magnitude less than the 0.19 PgC estimated by Moriarty and O'Brien (2013).
The CanESM5 total of 0.14 Pg is relatively close to the Moriarty estimate but implicitly includes
microzooplankton.

Surface chlorophyll and POC for CanESM5-CanOE and for ocean colour observational data are
shown in Figure 18 (POC in the model is the sum of phytoplankton, microzooplankton, and
detrital carbon). The observations have a lower limit for POC that is not present in the model
(~17 mgC m$^{-3}$), which is unsurprising given the processes neglected in the model, i.e., in regions
of very low chlorophyll there is still substantial dissolved organic carbon, bacteria that consume
it, and microzooplankton that consume the bacteria and produce particulate detritus. The
observational data show a fairly linear relationship at low concentrations, but with a curvature
that implies a greater phytoplankton fraction in more eutrophic environments (cf. Chisholm,
1992). The model, by contrast, shows a fairly linear relationship over the whole range of
concentrations. In other words, the phytoplankton share of POC is higher and more constant in
the model than in the observations. The living biomass (phytoplankton + microzooplankton)
fraction of total POC in CanOE is generally in excess of 50% (not shown), which is implausible



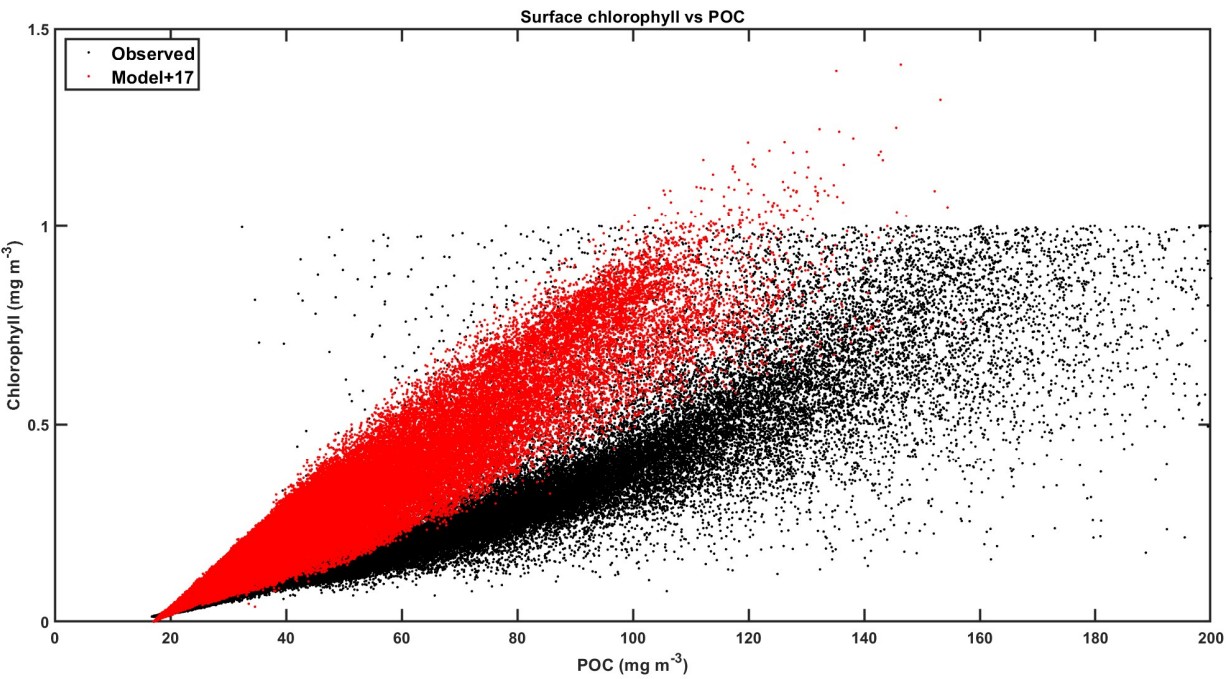

Figure 18 - Climatological surface particulate organic carbon (POC) vs. chlorophyll for CanESM5-CanOE (red) and observations (black). Data are for all ocean grid points (2x2° uniform global grid) for all months of the year where observational data are available. Model POC is offset 17 mg m$^{-3}$ for illustrative purposes.





for a real-world oceanic microbial community (e.g., Christian and Karl, 1994) but consistent
with the relatively low rates of export from the euphotic zone.

Export production for a range of CMIP6 models is shown in Figure 19a. CanESM5-CanOE is at
the low end of the range. Observations are not shown because the range of observational
estimates covers almost the entire range of the plot (e.g., Siegel et al., 2016). Note also that
CanESM5 export is quite a bit lower than in CanESM2, which is relatively high for CMIP5
models (not shown). The difference between CanESM2 and CanESM5 is attributable primarily
to different circulation, although the different initialization fields for nitrate might also play a
small role. The lower rate in CanESM5-CanOE is consistent with the above results regarding
plankton community structure (e.g., the concentration of detritus is generally low compared to
living biomass), as well as the lower sinking rate for small detritus. The latitudinal distribution of
export is shown in Figure 19b. CanESM5 shows very high export in the mid-latitudes of the
Southern Ocean, similar to CanESM2 (not shown). Both CanESM5 models show latitudinal
patterns consistent with the range of other models. CanESM5 has slightly greater export in the
equatorial zone; in both CanESM models the equatorial enrichment attenuates very rapidly with
latitude and the rates are low in the subtropics.

**3.5 Historical trends**

Cumulative ocean uptake of $CO_2$ is shown in Figure 20 for the historical experiment (1850-
2014). CanESM models are biased low relative to observation based estimates (~145 PgC, see
Friedlingstein et al., 2020) and the ensemble mean of other models (144 PgC, Figure 20), but fall





Figure 19 - (a) Global total export production (epc100) in PgC y⁻¹ (b) and zonal mean export production in molC m⁻² y⁻¹ according to selected CMIP6 models (mean for 1985-2014 of historical experiment). Thick red line represents CanESM5-CanOE, thick blue line CanESM5, thin grey lines individual CMIP6 models, and thick grey line the model ensemble mean (excluding CanESM5 and CanESM5-CanOE).







Figure 20 - Cumulative ocean uptake of carbon dioxide ($CO_2$) as anthropogenic dissolved inorganic carbon (AnthDIC) in PgC over the course of the historical experiment (1850-2014). Data are shown as successive five-year means. CMIP6 mean (thick grey line) indicates ensemble mean for CMIP6 models (thin grey lines) excluding CanESM5 (blue) and CanESM5-CanOE (red). An observation-based estimate of 145±20 PgC (Friedlingstein et al., 2020) is shown for nominal year 2014 (black).



well within the spread of CMIP6 models. CanESM5-CanOE has lower cumulative uptake than
CanESM5 by ~ 10 PgC. As the models were not fully equilibrated when the historical run was
launched, this difference does not necessarily arise from the biogeochemical model structure;
part of the difference can be attributed to differences in the spinup protocol (cf. Séférian et al.,
2016). The drift in the piControl experiment over the 165 years from the branching off of the
historical experiment is -10.5 PgC in CanESM5-CanOE and -5.9 PgC in CanESM5, so drift
accounts for about half (44%) of the difference in net ocean $CO_2$ uptake. The vertical distribution
of anthropogenic DIC is very similar between CanESM5 and CanESM5-CanOE (not shown).

The long-term trend in global total export production is shown in Figure 21. The model values
must be normalized in order to compare trends, since the differences among means are large
compared to the changes over the historical period (Figure 19). CanESM5 shows a greater
decline than most other CMIP6 models, while CanESM5-CanOE is more similar to non-
CanESM models. Such trends are difficult or impossible to meaningfully constrain with
observations, but the general expectation has been that export will decline somewhat due to
increasing stratification (e.g., Steinacher et al., 2010). The change in CanESM5 is geographically
widespread and not concentrated in a specific region or regions: export is maximal in the tropics
and the northern and southern mid-latitudes (Figure 19b) and declines over the historical period
in all of these regions (Supplementary Figure S1). In CanESM5-CanOE, export declines in the
same regions, but the magnitude of the change is smaller, and in the Southern Ocean increases
and decreases in different latitude bands largely offset each other.






Figure 21 - Change in export production (epc100) over the course of the historical experiment (1850-2014), normalized to the 1850-1900 mean. Data are shown as successive five-year means. Thick red line represents CanESM5-CanOE, thick blue line CanESM5, thin grey lines other CMIP6 models, and thick grey line the ensemble mean of non-CanESM models.



The trend in the volume of ocean water with $O_2$ concentration less than 6 or 60 mmol m$^{-3}$ is
shown in Figure 22. Again, the totals are normalized to a value close to the preindustrial, as the
differences among models are large (Figure 5). For the volume with <60 mmol m$^{-3}$, CanESM
models show relatively little change; in CanESM5 the volume actually declines slightly, while in
CanESM5-CanOE it increases, but the total change is <1% in each case. As with the baseline
volumes, the range among models is large, with one model showing an increase approaching
10% of the total volume estimated for WOA2013 (Figures 5b and 22b). For the volume with <6
mmol m$^{-3}$ (Figure 22a), CanESM models are among the most stable over time. In CanESM5, the
volume again declines, although this is within the range of internal variability. Again some
models show fairly large excursions, but in this case none shows a strong secular trend over the
last half-century.

**4. Discussion**

The development of CanOE was undertaken in response to some of the most severe limitations
of CanESM1/2, and in light of our collective experience. In addition to CMOC (Zahariev et al.,
2008), previous models developed by members of our group include Denman and Peña (1999;
2002), Christian et al. (2002a; 2002b), Christian (2005), and Denman et al. (2006). Christian et
al. (2002a) had a prognostic Fe cycle and multiple phytoplankton and zooplankton species, but
had fixed elemental ratios. Christian (2005) incorporated a cellular-regulation model, but only
for a single species and without Fe limitation. Christian (2005) had prognostic chlorophyll
whereas Denman and Pena (1999; 2002) and Christian et al. (2002a) used an irradiance-depend





Figure 22 - (a) Change in total ocean volume with oxygen ($O_2$) concentration less than (a) 6 mmol m$^{-3}$ and (b) 60 mmol m$^{-3}$ over the course of the historical experiment (1850-2014), normalized to the 1850-1870 mean. Data are shown as successive five-year means. Thick red line represents CanESM5-CanOE, thick blue line CanESM5, and thin grey lines other CMIP6 models.





ent diagnostic formulation. Christian et al. (2002a) used multiplicative (Franks et al., 1986) graz-
ing, which creates stability in predator-prey interactions but severely limits phytoplankton bio-
mass accumulation under nutrient-replete conditions.

One of the most important lessons from Christian et al. (2002a; 2002b) was that when a fixed
Fe/N ratio is employed, sensitivity to this parameter is extreme. Because Fe cell quotas are far
more variable than N, P, or Si quotas, treating this parameter as constant results in the specified
value influencing the overall solution far more than any other parameter. CanESM5-CanOE
largely succeeded in creating a prognostic Fe-N limitation model that produces HNLC conditions
in the expected regions (Figures 10, 11, 14, 15), although surface nitrate concentration is low rel-
ative to observation-based estimates in some cases. External Fe sources and scavenging parame-
terizations will be revisited and refined in future versions. We note that the aeolian mineral dust
deposition field employed here is derived from the CanESM atmosphere model; these processes
are not presently interactive but could be made so in the future.

A particular issue with CanESM2 was that extremely high concentrations of nitrate occurred un-
der the Eastern Boundary Current (EBC) upwelling regions. This error resulted from spreading
denitrification out over the ocean basin so that introduction of new fixed N from $N_2$ fixation
would balance denitrification losses within each vertical column, whereas in the real world deni-
trification is highly localized in the low oxygen environments under the EBCs. CanESM2 did not
include oxygen, but CanESM5 CMOC incorporates oxygen as a 'downstream' tracer that does
not feed back on other biogeochemical processes. The incorporation of a more process-based de-





nitrification parameterization in CanOE is independent of the many other processes that are pre-
sent in CanOE but not in CMOC: a CMOC-like model with prognostic denitrification is clearly
an option. We chose not to include explicit, oxygen-dependent denitrification in CanESM5 be-
cause we wanted to maintain a CMOC-based model as close to the CanESM2 version as possi-
ble, and because oxygen would not then be a downstream tracer that does not affect other pro-
cesses.

CanOE for the most part successfully reproduces the overall distribution of major tracers such as
nitrate, oxygen, DIC and alkalinity (and dFe, to the extent that its distribution is known). One
could argue that the gains made relative to CMOC are incremental. However, it is also important
to note that CanOE explicitly simulates important processes that are highly parameterized or
specified in CMOC. For example, the maintenance of HNLC regions is hardwired into CMOC
by specifying iron limitation as a function of the present-day observed distribution of surface ni-
trate. Both models show substantial gains in skill relative to CanESM2. These gains are similar
in the two models and in coupled or ocean only (with CanESM2 forcing) mode (not shown) and
are, therefore, attributable primarily to improvements in the ocean circulation model, although
differences in initialization and spinup may also play a small role (e.g., Séférian et al., 2016).

Plankton community structure in CanOE is somewhat biased toward high concentrations of phy-
toplankton, low concentrations of zooplankton and detritus, and low export (Sect. 3.4). In the de-
velopment phase, a fair number of experiments were conducted with various values of the graz-
ing rates and detritus sinking speeds. A wide range of values of these parameters was tested, with
no resulting improvement in the overall results. Possibly the detrital remineralization rates are



too high, although primary production is also on the low end of the CMIP6 range (not shown),
and would probably decline further if these rates were decreased. The model was designed
around the Armstrong (1994) hypothesis of 'supplementation' vs 'replacement', i.e., small phy-
toplankton and their grazers do not become much more abundant in more nutrient-rich environ-
ments but rather stay at about the same level and are joined by larger species that are absent in
more oligotrophic conditions (see also Chisholm, 1992; Landry et al., 1997; Friedrichs et al.,
2007). The results presented here suggest that this was partially achieved but further improve-
ment is possible (Figure 17).

As to whether the gains in skill with CanOE justify the extra computational cost, Taylor dia-
grams (Figures 4, 8, 9, and Supplementary Figure S2) show a modest but consistent gain across
variables and depths, especially for alkalinity at mid-depths (Supplementary Figure S2), for
which CanEM5 displays the least skill relative to other fields or depths. Other processes that are
highly parameterized in CMOC, such as calcification and $CaCO_3$ dissolution, were not addressed
in detail in this paper, but are an important factor in determining the subsurface distribution of
alkalinity. As noted above for maintenance of HNLC conditions, we emphasize that we are simu-
lating as an emergent property something that is parameterized in CMOC, and doing at least as
well in terms of model skill. As a general rule, the potential for improving skill and achieving
better results in novel environments (e.g., topographically complex regional domains like the
Arctic Ocean and the boreal marginal seas), is expected to be greater in less parameterized mod-
els (e.g., Friedrichs et al., 2007; Tesdal et al., 2016).





An updated version of CMOC with prognostic denitrification is clearly possible. However, for
the reasons discussed above, a prognostic Fe cycle with a fixed phytoplankton Fe/N remains
problematic, and the model would still have a single detritus sinking speed and remineralization
length scale. We are also developing CanOE for regional downscaling applications (Hayashida,
2018; Holdsworth et al., 2021), and it is likely that the simplification of having a single particle
sinking speed is not well suited to a domain with complex topography and prominent continental
shelf and slope. The number of tracers in CanOE is not particularly large compared with other
CMIP6 models. We expect to further refine CanOE and its parameterizations, evaluate it against
new and emerging ocean data sets (e.g., GEOTRACES, biogeochemical ARGO), and incremen-
tally improve CMOC (which we will maintain for a wide suite of physical-climate experiments
for which ocean biogeochemistry is not central to the purpose). For CMIP6, we chose to keep
CMOC as close to the CanESM2 version as possible. This strategy allows us to quantify how
much of the improvement in model skill is due to the physical circulation, which is in fact sub-
stantial (e.g., Figure 8). The CanESM terrestrial carbon model is also undergoing important new
developments (e.g., Asaadi and Arora, 2021) and we expect CanESM to continue to offer a cred-
ible contribution to global carbon cycle studies, as well as advancing regional downscaling and
impacts science.

*Code availability*. The full CanESM5 source code is publicly available at
gitlab.com/cccma/canesm; within this tree the CMOC/CanOE code can be found at
gitlab.com/cccma/cannemo/-/tree/v5.0.3/nemo/CONFIG/CCC_CANCPL_ORCA1_LIM_CMOC
or CCC_CANCPL_ORCA1_LIM_CANOE (last access: 21 September 2021). The version of the
code which can be used to produce all the simulations submitted to CMIP6, and described in this



paper, is tagged as v5.0.3 and has the associated DOI: https://doi.org/10.5281/zenodo.3251113
(Swart et al., 2019b).

*Data availability*. All CanESM5 simulations conducted for CMIP6, including those described in
this paper, are publicly available via the Earth System Grid Federation (ESGF). All observational
data and other CMIP6 model data used are publicly available.

*Author contributions.* Formulation of the overall research goals and aims: JRC, KLD, NS, NCS;
Implementation and testing of the model code: JRC, HH, AMH, WGL, OGJR, AES, NCS; Car-
rying out the experiments: JRC, WGL, OGJR, AES, NCS; Creation of the published work: JRC,
HH, AMH, AES, NS, NCS.

*Competing interests*. The authors declare that they have no conflict of interest.

*Disclaimer*. CanESM has been customized to run on the ECCC high-performance computer, and
a significant fraction of the software infrastructure used to run the model is specific to the indi-
vidual machines and architecture. While we publicly provide the code, we cannot provide any
support for migrating the model to different machines or architectures.

*Acknowledgments*. This work was made possible by the combined efforts of the CCCMa model
development team and computing support team. We thank the data contributors to and develop-
ers of the observation-based data products, the NASA ocean colour team, and all of the CMIP6



data contributors. The Python packages mocsy by Jim Orr and SkillMetrics by Peter Rochford
were invaluable tools in the analysis. William Merryfield and Andrew Ross made useful com-
ments on an earlier draft. This paper is dedicated to the memory of Mr. Fouad Majaess, who sup-
ported CCCMa supercomputer users for many years and passed away suddenly in 2020.

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
