# Peer review of "Ocean biogeochemistry in the Canadian Earth System Model version 5.0.3: CanESM5 and"

_Geoscientific Model Development, 2021_

## Author Response (AR1)

Overarching issues:

(1) We rationalized the terminology regarding our model naming convention, as it seemed to confuse several reviewers. CanESM5 and CanESM5-CanOE are official CMIP6 names. In the revised manuscript, we refer to our models by these official names as much as possible, and minimize references to "CMOC" and "CanOE" (which one could perhaps describe as our own in-house acronyms for our two biogeochemistry modules, although both are defined in publications). We have also removed most references to PISCES as these seemed to confuse the reviewers. Our model development strategy was to use the basic code structure of PISCES (i.e., how it organized biological processes into subroutines and source files), and insert into it our own process parameterizations. We tried to be explicit about the few cases were we had borrowed PISCES parameterizations and implemented slightly modified versions of them, but this seemed to confuse the reviewers who were sometimes unclear on whether we were talking about our own models or simulations employing PISCES. We have removed most references to PISCES, as the actual process models used are our own. We reference PISCES only where it is necessary to properly credit Aumont et al when we have used one of their parameterizations or a slightly modified version.

(2) We have expanded the Introduction (at the end) and Discussion (at the beginning) to address reviewer concerns that (a) we did not clearly explain why we structured the Results as we did (Reviewer #2), and (b) the Discussion launched immediately into the historical background and motivation for our model development choices without any meaningful discussion of the Results presented (Reviewer #1). We also moved some text out of the Introduction that Reviewer #2 thought more properly belonged in the Model Description.

(3) We have edited the Abstract in a number of places in accordance with the comments of Reviewer #2. This sort of paper covers a lot of ground and therefore we have to be selective about what is mentioned in the Abstract, but all of the points in the original Abstract did summarize conclusions drawn in the main text. In any case, we have edited the passages that the reviewer flagged, and tried to make sure that the meaning is clear and reflects the main text accurately.

(4) In accordance with the suggestion of Reviewer #3, we have made some direct comparisons of our modelled dissolved iron concentrations with GEOTRACES transect data. We chose GA-02 in the Atlantic because it was the most spatially extensive transect available. We show depth profiles from 47S to 47N. Mostly this confirms what we already knew from the other analyses presented: our model has a very low scavenging rate below 0.6 nM and a very high rate above, so that deep water concentrations are quite uniform and near-surface concentrations are biased low in high-deposition regions like the northern tropical Atlantic. For the most part, the model reproduces the observed concentrations quite well, given these known biases. What we learn from including this additional analysis is (a) the model is biased high in the Antarctic Bottom Water (which has quite low concentrations (0.2-0.4 nM), and (b) the seasonal biological drawdown in the mid-latitude North Atlantic is weak. The former is probably due mainly to the low scavenging rate at concentrations <0.6 nM, although it may also indicate a high bias in surface waters of the source region. The latter is probably related to the generally low rate of

export production (Figure 19) and the weak North Atlantic spring/summer bloom (Figures 16 and 17).

(5) There were several errors in the referencing of ancillary data sets and the offline calculation of derived carbon chemistry variables. We were using WOA2013 in some cases and WOA2018 in others (and incorrectly referenced only WOA2013 in the text), and an old beta version of GLODAPv2. All data products have been updated to the latest versions available and correctly referenced in the text. These differences are of no functional significance and have no effect on the conclusions. We also incorrectly calculated the pressure effect on $CaCO_3$ solubility. This affects the $\Omega_A$ and $\Omega_C$ estimates (Figures 6, 7 and 9), primarily at abyssal depths. The differences have little effect on the spatial patterns of $\Omega$ distribution and do not affect the conclusions drawn. The main impact is that the $\Omega_A$ estimates in the original Figure 6 were biased high (both models and observations) at the deepest depth shown (3500 m).

Response to Reviewer #1

This manuscript describes the incremental development of the marine biogeochemical component of the Canadian Earth System Model(s) v.5 and the contribution of these models to the 6th phase of the climate model intercomparison project (CMIP6).

My overall judgment is that this is a good technical publication, whose main aim is to describe the features of the new marine biochemistry component (CanOE) and provide useful insights on the historical simulations performed with both versions of CanESM5.

In commending the authors for their achievement, I would however point out that the discussion of the results obtained with CanESM5 and CanESM5-CanOE is not as detailed as one would expect and it should revised to better exploit the material presented in the results section (see detailed comment below).

I would also recommend the authors to revise the ending section of the manuscript with clearer perspective on future research directions and foreseen model development.

**We thank the reviewer for the constructive review. We have restructured the Discussion section in accordance with the comments of this and other reviewers.**

Specific comments

Section 3: In the results section the authors widely discuss about differences and biases in the comparison of the model outcomes with observations and CMIP6 multi model ensemble. So, it would be more effective to swap the Figures 2, 3, 6, 7 with the corresponding ones of the Supplementary Material (which directly show the differences against observations).

**We have swapped out the main and Supplementary figures as the reviewer suggests.**

L46: Is the NEMO model implemented on a T63 horizontal grid? I think this statement is not correct and should be modified to correctly address the ocean model configuration (likely ORCA1 grid). I suggest to add more details on the configuration and resolution of the different CanESM5 components in Section 2.

**No, the T63 grid is the atmosphere model. We have clarified this in the revised MS. As the reviewer suggests, the ocean is on the ORCA1 grid. This passage was deleted in any case (see Reviewer #2 comments). We have added more detail about the ESM as a whole to the beginning of Section2, as the reviewer suggests.**

L123: This aspect could be improved by adopting the SolveSAPHE solver (Munhoven, 2021 https://doi.org/10.5194/gmd-14-4225-2021) in the future development of the model

**We will consider the reviewer's suggestion for our future development. As the present contribution is part of CMIP6, we simply followed the OMIP-BGC protocols as outlined by Orr et al (2017).**

L127: I think it should be stated in here that carbon chemistry variables are computed offline instead of discovering it at L360

**This section is model description. Here we are describing how the carbon chemistry is solved inside the model. Carbon chemistry caclulations for the plots in this paper were done offline from published DIC and alkalinity, but that is a separate issue. Section 2.6 Ancillary Data seems to us the appropriate place to state what was done for the offline carbon chemistry caclulations.**

L360: GLODAP is a versioned dataset. It would be clearer to refer to it as GLODAPv2.

**We were a bit careless in our earlier calculations: some of the data sets used were not the versions stated. All of the figures have been remade and the latest versions of all gridded data sets used. The differences are of no functional significance and do not alter any of the conclusions.**

L360-363: How different is the carbon chemistry obtained with the online computation? If applicable, I suggest authors to detail this aspect to support the offline approach.

**We have done this calculation many times with many data sets. The differences are negligible from the perspective of the kind of global-scale analysis with which we are currently concerned.**

L382: A more substantiated explanation should be provided to explain the use of such a coarse horizontal sampling (2°x2°) of CanESM2 and CMIP6 datasets.

**Again we are primarily concerned with documenting the global-scale distributions of major tracers. For this purpose, whether the data are regridded at 1deg or 2deg makes little difference. For example, when a 1x1 grid is used, none of the correlation coefficients for CanESM5 vs observed oxygen on the six depth levels shown in Figure 4 changes by more than 0.005 (max 0.0028, mean 0.0011). This is noted in the revised MS.**

L401: The CMIP6 multimodel ensemble data are treated here (and in the following paragraphs) as a "single model" results, but I think that authors are missing the opportunity to exploit this information to better characterize CanESM5 performance in the broad CMIP context

**We chose this particular method of presentation because in this case presenting all of the individual CMIP6 models seemed to us to be potentially overwhelming the reader with questionably relevant information. We have modified the text to try to address the reviewer's concern.**

L410: A description of the Oxygen Minimum Zones spatial patterns would a good complement to this paragraph.

**done**

Figure 5: Axis labels should be increased in size to make them easily readable

**done**

L460-464: The observation-based Aragonite saturation state is here recomputed using GLODAPv2 and WOA2013 data instead of using the original field made available within the GLODAPv2 dataset. The rationale for this choice should be specifically addressed.

**We do all carbon chemistry calculations offline to make sure they are done in a consistent way across models and observations.**

L484-485: I don't think that the conclusion made by Lambert and Boer (2001) in the analysis of atmospheric fields (air temperature, precipitation, sea level pressure) from CMIP1 exercise can be extended in such a way to the DIC, and more generally, to any ocean biogeochemistry.

**There are differences of opinion about the appropriateness of citing older vs more recent literature. We prefer to cite the reference that first (to our knowledge) articulated an idea. But this idea has proved over time to be quite robust, and it is generally true for all sorts of fields including ocean biogeochemistry fields (see e.g., Figure 22 of Chapter 5 of the AR6 (WG1) report). It can also be deduced from the data shown in the paper itself, which include both ensemble means and individual models for several example ocean biogeochemistry fields (e.g., Figures 4, 8, 9, and S2), as noted by reviewer #2.**

L515: Figure 11b could be moved to Supplementary material.

**Figure 11b has been moved to Supplementary as requested.**

L526: The comparison of dFe observations with different models outcome in Figure S4d could be improved by reporting also the tendency lines of each model along with the ideal fit (1:1) black line.

**Yes good idea. This was added.**

Besides, these results could be further discussed in the light of the findings from Seferian et al. (2020, https://doi.org/10.1007/s40641-020-00160-0)

**This has been added to the discussion of Fe model skill in the Supplementary material. Unfortunately Seferian et al. did not include any analysis of CanESM5-CanOE dissolved iron distribution. We provided these data to Dr. Seferian but may have done so too late. That paper was prepared under substantial time pressure to meet the AR6 submission deadline.**

Figure 14: revise caption by adding "CanESM5 is not included because it does not have prognostic iron"

**done**

L568: here it would be interesting to specify which are the two models that do not fall along the spectrum.

**done**

L574 and L577: Use 260°E instead of 100°W, coherently with the Longitude units used in Fig. 15.

**The figure labels were modified to show E and W longitude.**

L577: It would be useful to have this "not shown" figure in the Supplementary material

**Not shown in this context indicates analysis of a number of data sets, all of which are in the public domain. The point at issue is relatively trivial: whether this particular local maximum in surface nitrate concentration is due to undersampling. Such localized maxima certainly do exist in this data product, but in this case the maximum is associated with equatorial upwelling at the longitude where the thermocline is nearest the surface. We have reworked the wording (there is no longer a "not shown"), and added a literature reference that shows that the flow is strongly divergent at this location.**

Section 3.4: Differently from the previous ones, this part is largely intertwined with comments on results that better fits the discussion section.

**We have addressed this as part of a more general restructuring as per the reviewer's general comments above.**

L587: Add reference to Tesdal et al. (2016)

**The reference is given in the figure caption. It seems like overkill to repeat in the text especially given that the reference given is for a climatology of chlorophyll concentration. The text refers to biomass and the figure caption explains how biomass was estimated from chlorophyll.**

Figure15: Authors should consider to add a shaded area showing, e.g., the min-max range obtained from the CMIP6 model ensemble and include some considerations with respect to CanESM5.

**We have tried several different versions of this, but we don't think it adds very much. The range is very large, as the reader can deduce from other figures shown in the paper. This is discussed in the revised MS.**

Figure 16: Axis labels should be increased in size to make them easily readable

**done**

L601-603: The expected behavior of phytoplankton size distribution is clearly visible only in subarctic regions, while in the North Atlantic the monthly variability is rather similar between small and large classes.

**The reviewer may have mistaken the Total line for one of the size classes. The figure clearly shows that the amplitude of the seasonal cycle is greater for large phytoplankton which exceed 50% of the total only in summer and fall to near 0 in winter.**

Figure 17: The supplementary Table S4 could be easily replaced with a map illustrating in a more straightforward way the location and extent of selected marine regions.

**We think the Table is necessary so that the reader can see the actual numbers for the region bounds, but have included a map as well.**

L655: I guess it should read as "... with the range of other CMIP6 models."

**CMIP6 added**

L671-673: It could be useful to address in a dedicated table the residual drifts of the piControl simulation for the CO2 uptake and also the other biogeochemical variables presented in the previous sections.

**This is a good suggestion and overlaps with one made by reviewer #2. We added a Supplementary Table to illustrate the magnitude of historical trends relative to drift.**

Section 4: There are several parts of the discussion section that are not fitting the real purpose. For example, L704-716 describes the evolution of the model which in my option should pertain to the introduction. L729-741 focuses on the differences between CanESM2 and CanESM5 formulations which was already stated previously in the manuscript. Lastly, in many points the authors refer to not shown figures that is not helpful to the discussion. I suggest to revise the entire section by tackling the various outcomes of the result sections, which is very rich in content and material.

**The Discussion section has been restructured according to the reviewer's comments. However, we disagree that the historical background belongs in the Introduction, which is already quite long. We have retained this material in the Discussion, following an extensive discussion of the Results presented as requested by the reviewer.**

L771: Typo in the model name "CanESM5"

**fixed**

L774-776: This sentence is not clear

**Sentence has been restructured to make the meaning clearer.**

L792-794: The impact of the different ocean circulation between CanESM2 and CanESM5 is supported only by the comparison of DIC (Fig.8) and the one "not shown" at Line 750. I think that this part should be better supported (maybe with some additional analysis on other variables) to robustly claim that only ocean circulation is responsible for the observed differences between the two model versions.

**We have expanded our discussion of the relative roles of circulation and biogeochemistry in a number of places in response to comments by several reviewers. The text here refers to "e.g., Figure 8" but the assertion is also substantiated by Figure 9 and Figure S2. The text has been modified to reflect this. That there is no similar comparison for O2 is unfortunate; there are no CanESM2 O2 data because at the time we were under a lot of pressure to keep the number of tracers to a minimum. We chose not to include in the Supplementary material the geographic distribution of DIC referred to as "not shown" on 489 because all of the relevant data are in the public domain and the interested reader can easily verify this.**

Response to Reviewer #2

First of all, I would just want to make clear that I think that this sort of publication is valuable for documenting in detail the performance of major components of ESMs, especially where, as here, the authors make an effort to cover the model end-to-end, as well as compare with peer models. Given the broad span of tracers, processes and geographical / historical patterns, there is no natural end to where such analysis should begin or stop. That said, I have a number of major criticisms of the current draft of this manuscript:

While the focus is on marine biogeochemistry, it is remiss not to include information about the performance of the physical model that underpins this. The reader has no information on how well this latter model performs in terms of surface properties, mixing and ventilation and interior circulation. This need not be exhaustive, but some material on this (even if only summarising from another evaluation manuscript) seems important. Especially where this has a potential bearing on biogeochemistry performance.

**We thank the reviewer for a thorough and constructive review.**

**A basic evaluation of the physical ocean model was presented in the overview paper by Swart et al. (2019). This includes comparisons to observations of SST, SSS, SSH, zonal mean T+S, sea ice extent and volume (seasonal cycle), and spatial distribution of sea ice in March and September. The MOC is shown as depth-latitude plots (Atlantic, Pacific, global) as is the integrated meridional heat transport (with observation based estimates at a few discrete latitudes). We believe that this analysis is adequate for the present purpose, but we did not do a good job of drawing the reader's attention to it. This has been corrected in the revised MS. In addition, we believe that some of our analyses (e.g., of oxygen distribution) are useful diagnostics of the performance of both the physical and the biogeochemical models. This was not presented clearly in the initial draft, and is addressed in the revised manuscript according to concerns raised by this and other reviewers.**

The evaluation itself seems almost arbitrary in its choice of targets and, in particular, the order in which these are introduced and discussed. For me, oxygen and carbonate chemistry parameters are essentially "downstream" of the main drivers of biogeochemistry in the ocean. Nutrient cycles, productivity and carbon / alkalinity seem much more important to first-order patterns. And this leads to oddities in the manuscript, for instance where patterns of oxygen biases, likely due to "upstream" biases in production, are discussed prior to anything about these likely sources. Also, the focus on export production instead of primary production is rather odd.

**A wholesale restructuring of the Results would be difficult to achieve within the time frame available. We have tried to accommodate the reviewer's perspective as far as possible, and have added some text to more clearly explain why we chose to structure the paper in this way.**

**A key objective of the ESM intercomparisons is to evaluate the effects of climate change on the distributions of major tracers like oxygen, DIC, alkalinity and nitrate. And the major tracers are better observed: gridded data sets are available over the full ocean depth, which is important for evaluating models that take thousands of years to spin up. For biogenic**

**particulates, satellite surface chlorophyll and POC are the only reliable global data sets, and even these have limited utility for validating coarse resolution global models (e.g., very high chlorophyll in coastal regions, associated with processes not resolved by the model). Realism with respect to plankton distributions and productivity are necessarily limited compared to ocean-only hindcast models or higher-resolution regional models. The ESMs are needed to provide boundary conditions for climate downscaling experiments with such models, and the requirement for such boundary conditions is limited to the slowly-evolving major tracers.**

Latterly, the manuscript looks at historical trends in anthro CO2 uptake, export production and ocean anoxia. While these are valuable to look at, the manuscript offers little by way of explanation for them. These aren't easy issues to tackle, and are probably beyond the scope of this manuscript, but it seems remiss not to include analysis. For instance, it might be informative to show whether there are spatial components to these trends, or links to physical phenomena. In the specific case of export production, the authors allude to long-known issues with increasing stratification and decreasing primary production, but show no evidence of either (further, I think the physics might be the same in both models).

**We have tried to address this in concert with other reviewer comments, particularly with regard to the structuring of the various sections (I-M-R-D), and more discussion of the physical mechanisms underlying some of the results presented.**

Finally, on the model naming side, while the manuscript eventually settles down, it is confusing at first about the identities and compositions of the CanESM models under consideration. The introduction and, especially, the abstract are messy on this front, and will likely confuse readers. A clear and simple statement

**The ending of this comment seems to be missing, but it is fairly clear what the reviewer is trying to say. This issue was raised by several reviewers. We acknowledge our carelessness in this respect, and have tried to make the terminology consistent throughout the revised MS.**

In addition, I have a number of specific comments on details of the manuscript and include these below. I should add that some of these reflect my own style / presentation preferences, and the authors should not feel obliged to address these if they disagree.

Overall, while I appreciated much of this manuscript, I judge that it requires major revisions before it can be accepted.

Specific comments:

Ln. 19-22: This opening is confusing; which models are being examined here?; if there are more than one, and it sounds like there are, say this in the opening sentence

**We apologize for the vague wording. We have clarified it in the revised MS.**

Ln. 24: a brief mention of the aspects of the model being intercompared would be useful (e.g. nutrients, carbon, productivity, etc.)

**added**

Ln. 33: re: export decline - why?; is this explored in the manuscript?

**This result was shown in the paper (line 677-678), although the underlying mechanisms were not explored in detail.**

Ln. 33-35: re: plankton - is this worth mentioning in the abstract?; it's without context and isn't clear whether it's something that can be compared to observations

Ln. 35-36: re: phytoplankton - is this worth mentioning in the abstract?; again, there's just no context here - e.g. is this different between the different model versions considered?

**These results can not be directly compared to observations, but we think they are important in terms of understanding how the dynamics of the plankton community work in CanESM5 and CanESM5-CanOE. We have reworded the text slightly to clarify this.**

Ln. 39: if you're going to put the specific numbers in the abstract, you should also include some reference to the observational estimate (or range)

**done**

Ln. 39: re: anthro CO2 uptake differences - why?; is this explored in the manuscript?

**Again, this result was shown in the paper, although the underlying mechanisms were not explored in detail. As this is a key, and often cited, diagnostic of CMIP model performance, and is detailed for CanESM5 in the CanESM5 overview paper by Swart et al. (2019), we think the relative magnitude of uptake in CanESM5 and CanESM5-CanOE is of interest to the reader.**

Ln. 41: some of this material seems more appropriate for the methods section than the introduction

**In accordance with the reviewer's comment, we have moved some of the more detailed passages into the Methods.**

Ln. 41: more generally, the manuscript makes some specific choices on the model components and processes analysed, but does not articulate the science cases for these; the introduction is the place for doing this

**We added a section at the end of the Introduction to address the reviewer's general comment about the choice of data fields to analyze and the order in which they are presented.**

Ln. 48: just out of interest, why not v3.6_stable?; that's a common CMIP6 configuration; (although having a different version isn't necessarily an issue - it's more variety in the CMIP6 ensemble)

**We worked from what was available at the time when we began adapting NEMO for our ocean model. As some in-house parameterizations of processes in the physical ocean were implemented and coupling to the atmosphere was underway, there was not time to upgrade the ocean to NEMO3.6 when it became available.**

Ln. 50: this is PISCES-v2?

**No, we worked from PISCES-v1 which was what we took as a point of departure in NEMO3.4. The relevant processes did not change between PISCES-v1 and v2, and the description in Aumont et al, 2015 is an accurate description of the process model we used. We have removed most references to PISCES from the text as these seemed to confuse the reviewers. Our model development strategy was to use the basic code structure of PISCES (i.e., how it organized biological processes into subroutines and source files), and insert into it our own process parameterizations. We tried to be explicit about the few cases where we had borrowed PISCES parameterizations and implemented slightly modified versions of them. We have removed most references to PISCES, as the actual process models used are our own. We reference PISCES only where it is necessary to properly credit Aumont et al when we have used one of their parameterizations or a slightly modified version.**

Ln. 66: how does tracer number relate to compute cost?

**The total computational cost scales approximately linearly with the number of tracers which (as discussed below) is one of the reasons for implementing the additional OMIP-BGC tracers in CanESM5 rather than CanESM5-CanOE. This paragraph has been substantially rewritten.**

Ln. 66: the relationship between CMOC and the models here is unclear; how does this relate to CanESM1, 2 and 5?; this should be unambiguous

**We apologize for the confusion in the terminology. We have tried to make it clear and consistent through the revised MS. We refer to CMOC in reference to CanESM1, 2 and 5 because the biological process models are identical in each case.**

Ln. 69: re: CFCs - if the physics is the same between the models, you should only need to run the CFCs and SF6 in a single simulation anyway; they are non-interactive with the BGC

**Yes, that is exactly what we meant. We ran these tracers in CanESM5 because the cost of CanESM5-CanOE with its much larger suite of tracers is already large. This paragraph has been substantially rewritten.**

Ln. 71: re: "prohibitively expensive" - this isn't clear as many of these tracers are single tracers; or are you thinking that you might want to duplicate all of your BGC model tracers to have parallel reservoirs of natural C and C14?

**No, we simply meant that we were trying to avoid the incremental cost of these additional tracers (even if there are only a few) on top of the already large cost of CanOE.**

Ln. 74: does CMOC not have a version number?; seems odd as the version here is different (via O2) from earlier versions; not necessarily meaningfully (since O2 is a "downstream" tracer), but this does suggest at least a code difference

Ln. 74: there would be much less risk of confusion here if the different versions of CMOC here were identified with version numbers

**Yes this would be good, but assigning version numbers (and placing the source code in the public domain) is a recent innovation. In the early years, our version control was rather ad hoc. All published simulations with CanESM5 (CMOC) have a version number because CMOC is the biogeochemistry model in CanESM5, but CanESM1 and CanESM2 did not.**

Ln. 80-81: this specification of Fe limitation as being calculated from surface nitrate could do with a bit of justification; presumably this has been given before in a previous outline of the model; please include this here

**Yes this is described in detail in Zahariev et al (2008), but a brief explanation has been added.**

Ln. 95: re: 100% burial - is this true regardless of the saturation state of the seafloor?; i.e. above / below CCD

**Yes this is the case, because when we originally developed CMOC we did not solve the carbon chemistry in the subsurface layers and were primarily concerned with global conservation of alkalinity. One of the things we have tried to achieve with CanOE is to make burial dependent on saturation state, although in the current version we implemented only a rather simplistic representation of this. This passage has been rewritten to make sure there is no ambiguity about which model we are talking about in each case.**

Ln. 99: maybe referring to CanESM5-CMOC might be better than CanESM5 on its own

**Again, we apologize for the confusing terminology in the original draft. We sometimes use the terms as we use them among ourselves, and were a bit careless about how they were used in the initial submission. Referring to CanESM5-CMOC is not appropriate because**

**CanESM5 (which has CMOC as its biogeochemistry) is an official CMIP6 name. We submitted data from two models to CMIP6: CanESM5 and CanESM5-CanOE. So we can not change these names at this stage. In the revised MS we largely refer to CanESM5 and CanESM5-CanOE and have kept references to CMOC to a minimum.**

Ln. 105: I would have expected some sort of summary about the physical ocean (and sea-ice) model used here; including resolution and major options selected; especially as the common NEMO configuration v3.6_stable has not been used here

**As noted above, a basic evaluation of the physical ocean model was presented in the overview paper by Swart et al. (2019). We have revised the text to make this clear to the reader.**

Ln. 117: modified from what?; do you mean from previous versions of this model, or from elements of PISCES?; the latter is implied, but it would be useful to have a "... modified from corresponding PISCES components to varying degrees."

**Sorry, this was vague. It has been rewritten for clarity. As noted above (L50), our model development strategy was to use the basic code structure of PISCES and insert into it our own process parameterizations. We have removed references to PISCES except where it is necessary to credit Aumont et al when we have used one of their parameterizations.**

Ln. 118-119: and now we have another name for the model, NEMO-CMOC; some standardisation of naming would be helpful

**Again, we apologize for the careless terminology. We refer to NEMO-CMOC to distinguish it from the original CMOC which had the same process parameterizations but an entirely different ocean model. The terminology has been cleaned up and made consistent through the paper.**

Ln. 121: does the model use the preferred carbonate chemistry of Orr et al. (2017)?; MOCSY

**Yes. We do not use mocsy per se but all of the equilibrium constants etc. are identical to mocsy with the OMIP-BGC specified options. We took the original PISCESv1 carbon chemistry and replaced whatever was not consistent with Orr et al.**

Ln. 126: this is ambiguous; previously it's implied that all calcite reaching the seafloor is buried, whereas this line implies otherwise

**One refers to CMOC and one to CanOE. This passage has been rewritten to make sure there is no ambiguity about which model we are talking about in each case.**

Ln. 131: re: phytoplankton functional types - ... which could be named here

**Modified to specify large and small phytoplankton**

Ln. 130-137: this summary paragraph of the model focuses on the phytoplankton, and doesn't mention other components, e.g. zooplankton; that tends to imply they're the same as before; anyway, this would be a good place to introduce other elements

**Yes, good point. This paragraph has been modified to emphasize that CanOE has multiple size classes of zooplankton and detritus as well as phytoplankton.**

Ln. 142: are we getting an explanation of what these size categories are meant to represent?; e.g. prokaryotic vs. eukaryotic

**We do not think it is appropriate to associate the size classes with specific taxa with only two groups. There are places in the ocean where the small phytoplankton (strong grazer control and limited biomass range) are overwhelmingly (mostly prokaryotic) picophytoplankton, and regions where they are predominantly eukaryotic nanoplankton.**

Figure 1: as already noted, I'd suggest properly introducing all of the elements of this model in section 2

**Done as noted above**

Figure 1: for ease of comparison, and especially as it is not described here, it might be an idea to include the corresponding schematic of CMOC;

**We have added the CMOC schematic to the Supplementary material.**

Table 1: might it be worth noting where parameters have a corresponding parameter in CMOC, and whether the values are the same?; some model processes certainly overlap

**Yes, but there are relatively few parameters that fit in this category, and it leads to a proliferation of footnotes (e.g., to indicate that K_NiX in CanOE is only approximately equivalent to K_DIN in CMOC). We believe that the interested reader can easily work this out for him/herself.**

Table 1: re: parameter k_Ca - does CaCO3 dissolve above the CCD?

**Yes. Sinking CaCO3 is subject to first-order dissolution at all depths. This was stated on Line 288. It is well documented that dissolution above the saturation horizon occurs (e.g., Milliman et al 1999, DSR I 46: 1653), although clearly there is a saturation-state dependence of the rate (e.g., 10.1002/2013GB004619), and the mechanisms are not entirely understood. The basic version of CanOE does not include saturation-state dependence of water-column dissolution because we wanted a 'base' version to branch out from to experiment with ocean acidification feedbacks to the carbon cycle.**

Table 1: re: parameter K_NH4ox - might it be better to describe K_NH4ox as a maximum nitrification rate, which is then diminished by an irradiance function with a half-saturation, K_E

**OK. Possibly just 'Nitrification rate constant' is more appropriate; it's really a rate constant, not a rate. (We also changed the K to lowercase to make it consistent with equation 20.)**

Table 1: re: parameter K_NO3 - presumably N2-fixation occurs in ignorance of PO4 availability?

**Yes. DNF is dependent on light, temperature, dissolved iron, and DIN. We did not include a parameterization of PO4 limitation (although we have developed one on an experimental basis).**

Ln. 160: explain what's going on here with NH4 and NO3; does this functional form have a source describing it?

**This is a commonly used formulation for NH4 inhibition of NO3 uptake, although possibly written in a unfamiliar way. It reduces to the formulation based on noncompetitive inhibition (Frost and Franzen 1992 MEPS 83: 291; Hood and Christian 2008 in Capone et al eds "Nitrogen in the marine environment").**

Ln. 169: is E irradiance?

**Yes. We added this to the text. Sorry.**

Ln. 175: re: C_XS - this term needs expansion (or a reference to a later equation number if it appears below); I'm uncertain what you mean by this, or what ecological process it's meant to represent; equation 16a seems to be the right one

**Added reference to equation 16a where the symbol first appears.**

Ln. 180: "excessively low" in a model stability sense?; or is there an actual observed threshold here?

**This is simply a device used to prevent biomass from declining to levels far below the 'seed' population required for realistic biomass to accumulate in spring-summer, even under the most favourable growth conditions. We do not know exactly what limits the losses in the real world, but we know that something does. In any case phytoplankton linear mortality terms are a very inexact representation of any real process. First-order mortality leads to biomass declining to levels that make it impossible for the population to meaningfully recover in the brief Arctic summer. We have expanded the text here a bit to make this clear.**

Ln. 204-205: how equations 13a and 13b fit into equation 11b is unclear, especially as equation 11b refers only to G_L, which seems to be calculated in equation 12b; the latter point also occurs for small zooplankton

**Sorry there was a typo in Equation 13b; this may have been the source of the confusion. This has been corrected.**

Ln. 208-216: might the clarity of this section be improved by the addition of a diagram that quantitatively illustrates the scale of excess C, N (and possibly Fe) over a span of intake C:N?

Table 2: this kind-of answers my point before about a diagram, although a diagram might still be better (if more difficult to create)

**We prefer a table. It presents the critical information in a concise way.**

Ln. 241: has the impact of forcing a common zooplankton C:N on detritus compared to dynamic C:N in phytoplankton been explored at all?; does this mean that the majority source of both detritus classes is zooplankton?; it seems odd to make a fuss about C:N in phytoplankton only to entirely overlook the C:N of the more heterogeneously-sourced detritus component

**No, detritus originates from both phytoplankton and zooplankton, and under some conditions it comes predominantly from phytoplankton. It is true that the recycling of 'excess' phytoplankton C or N into the dissolved pool is probably unrealistic in some cases, but it is necessary to maintain mass conservation. Not including variable C/N in detritus was a purely pragmatic choice that was made to limit the number of tracers.**

Ln. 263: ah-ha; E = irradiance

**addressed above**

Ln. 265: does the absence of PO4 in the model cause any problems for this N2-fixation scheme?; low NO3 is often associated with low PO4

**Yes but there was no realistic way to parameterize this. For carbon chemistry we followed Orr et al at assumed that the PO4 contribution to alkalinity can be estimated as DIN/16. But N2 fixation tends to be associated with large departures from the N/P Redfield ratio that occur under extreme oligotrophic conditions. N2 fixation models are still at a rather primitive stage of development. CanOE is a step forward over CMOC in that it at least includes Fe limitation, whereas in CMOC N2 fixation tends to grow without bound in a warming ocean (Riche and Christian 2018).**

Ln. 284-285: does this scheme produce large-scale spatial patterns in calcite production that match the general high-equatorial, low-polar pattern?

**Yes, calcification occurs predominantly in the low latitudes in CanOE. But we believe that most CMIP5 models (including CMOC) overestimated the degree to which rain ratios decline with latitude or temperature (see Eq 13 and Figure 1 of Zahariev et al (2008)). Honjo et al (2010, Progr. Oceanogr. 85: 137) show that Arctic rain ratios are similar to the global mean, although in some cases they are very low (probably associated with diatom blooms). Some of the worst misfits of CMOC with the regional mean rain ratios estimated**

**by Sarmiento et al (2002) are due to the (probably excessively) strong dependence of rain ratio on SST (e.g., the subarctic Pacific).**

Ln. 290: again, it's implied earlier that 100% of calcite is buried, but this suggests otherwise

**No. 100% of calcite is buried in CMOC, but not in CanOE. The loss of alkalinity to burial is treated in the same way (reintroduced at surface in the same vertical column). The text has been revised to make this clear.**

Ln. 291-292: is this localised spatially?; i.e. loss at the seafloor is added at the surface immediately above

**see previous point**

Ln. 296: some expansion here on the precise links between processes would be helpful; e.g. NO3 vs. NH4

**added**

Ln. 317-318: this is a little paradoxical; the closer a seafloor tile is to sources of O2 (surface productivity and the atmosphere), the less oxygenated the sediments; this presumably reflects the supply of organic matter to the seafloor and the resulting oxygen demand; if this is the logic, make this clear

**Yes, it is based on the greater organic deposition and presence of reducing sediments at shelf depths. The text has been expanded a bit to make this clear.**

Ln. 328: this also implies that particles can scavenge iron continuously without saturation; I don't imagine this is a problem, but it might make the model's behaviour in areas dominated by slow or fast sinking detritus interestingly different

**Interesting point. No we did not consider such 'saturation'. But it is unlikely to be a major factor. Possibly a topic for a future experiment, although on the list of oversimplifications in our scavenging model it probably ranks fairly low.**

Ln. 332-340: sensible; I like this

**Thanks**

Ln. 342-352: this could be clearer and sourced to relevant work on the topic; Wolf-Gladrow et al. (2007) (which you cite earlier) suggest +1 ALK for N2-fixation to NH4+, -2 ALK for NH4+ to NO3-, and +1 ALK for denitrification of NO3-; here, assuming N2-fixation goes to NO3-, this implies -1 ALK for N2-fixation and +1 ALK for denitrification; anyway, the text here is ambiguous, and should be straightened out and sourced

**All of the sources and sinks of alkalinity associated with N cycle processes are detailed in Table S2. In an earlier draft this table was included in the main text, but we were afraid that reviewers would dismiss it as a reiteration of well-known information. This paragraph is important to make clear to the reader how alkalinity is conserved globally given that both N2 fixation and denitrification are prognostic. The text has been revised to make sure this is clear.**

Ln. 354: links for the data?; and access dates; some of these products are revised periodically

 **See next point**

Ln. 360: this is GLODAPv2

**We were a bit careless in our earlier calculations: some of the data sets used were not the versions stated. All of the figures have been remade and the latest versions of all gridded data sets used. The differences are of no functional significance and do not alter any of the conclusions.**
Ln. 360: which offline carbonate chemistry calculations are needed?; GLODAPv2 includes pH

**We do all carbon chemistry calculations offline to make sure that it is done in a consistent way across models and observations, because the main purpose is to assess the modelled distributions of DIC and alkalinity, which are the primary determinants of e.g., $\Omega\_A$. We have tested this many times (e.g., offline $\Omega\_A$ vs $\Omega\_A$ output from the model) and the differences are minor.**

Ln. 363: rephrase to "... were used for the absent tracers, phosphate ..."

**The text states accurately what was done. We do not think the reference to absent tracers is appropriate because this calculation was applied consistently across models (some of which have these tracers). Again, this is key to our decision to do C chemistry calculations offline: not all models have the same suite of nutrients, but the 3D distribution of e.g., $\Omega\_A$ is primarily determined by DIC and alkalinity.**

Ln. 382: why 2x2 degree?; the 33 levels is more understandable

**We regridded all data to a uniform grid for easy comparison across models. For the global scale patterns that we are primarily concerned with here, there is no meaningful difference between gridding at 2x2 or 1x1 degree. For example, when a 1x1 grid is used, none of the correlation coefficients for CanESM5 vs observed oxygen on the six depth levels shown in Figure 4 changes by more than 0.005 (max 0.0028, mean 0.0011). This is stated in the revised MS.**

Ln. 382: how regridded?; linear, nearest neighbour, etc.?

**This is specified in the revised MS.**

Ln. 383: technically, GLODAP follows WOA (which did this vertical grid first)

**Clarified in the revised MS.**

Ln. 386: I think ignoring variability across the CanESM5 ensemble is not an unreasonable assumption, but it might be useful to support it for this particular model with some evidence; e.g. a plot of some key property (e.g. NPP, CO2 flux, SST, etc.) across the ensemble for, say, the decades of interest here; this could be put in supplementary if it breaks the flow

**We added a Supplementary table (S4) that shows the correlation of modelled and observed DIC at six different depths for five CanESM5 ensemble members. It shows that the arbitrary selection of a specific ensemble member has little effect on the results.**

Ln. 394: it would be helpful to name (and source to descriptions / evaluations) the CMIP6 models used in this analysis within this section

**added**

Ln. 396: while it may have been done elsewhere, some sort of outline of the performance of the physics model seems necessary to me; even if it's cursory and largely points to this other work; if there is no other work, some expansion would be useful; things like surface physics (incl. mixing), sea-ice, major circulation (AMOC, Drake), MOC would be of interest

**This is addressed in the response to the reviewer's general comments above.**

Ln. 398: please be clear why you're starting with oxygen; in most models it's largely slaved to other more dynamic model processes and tracers (which are, in turn, strongly influenced by physics processes); if there's a good reason why you're looking at it first, make it clear

**The reasons for structuring the paper in this way are now clearly stated.**

Ln. 400: why these depths?; including a more abyssal depth might hint at circulation issues

**The depths were chosen to allow the reader to assess the models' representation of the spatial distribution of low-oxygen waters, particularly in the Pacific. These regions and depths contribute disproportionately to the global model-data misfit, and we think it is important for readers to understand how well, or poorly, the models are representing the underlying processes.**

**Abyssal depths are much less diagnostic of these specific processes (formation and maintenance of the oxygen minimum zones) although, as the reviewer notes, there are issues with ventilation of the deep ocean in some models. This issue is discussed briefly in the revised MS, but really deserves a paper all by itself.**

Ln. 401: for a number of reasons, I would not expect to MEM to be a good comparison; do you know how it compares to observations relative to the performance of its component models?

**The idea that a MEM outperforms individual models (Lambert and Boer, 2001) has proved over time to be quite robust, and it is generally true for all sorts of fields including ocean biogeochemistry fields (see e.g., Figure 22 of Chapter 5 of the AR6 (WG1) report). It can also be deduced from the data shown in the paper itself as noted by the reviewer below (comment on Figure 8).**

Figure 2: with fewer colours in this scale, it would be easier to discern differences between the models

**We have narrowed the ranges of all of the colour scales as much as possible.**

Ln. 422: you're inferring these as "circulation features" but haven't reported on your model's circulation at all (e.g. MOC)

**As noted above and in the responses to the other reviewers, we have tried to be more clear and specific in our references to ocean circulation processes and refer specifically to the analysis presented by Swart et al (2019). In this particular case, the specific circulation processes referred to were clearly stated in the text.**

 Ln. 428: "the ensemble mean" = "MEM"

**We were careless in our terminology here and have made it consistent throughout the revised MS.**

Ln. 433: it's old-fashioned of me, but would a profile of O2 further assist here?; possibly not given its spatial heterogeneity, but a series of vertical Taylor diagram slices is a little hard to take in!

**We accept that there is a bit of an 'information overload' factor here, but we believe that the multiple Taylor diagrams for different depths are a quite powerful way of visualizing model skill. Clearly it only tells you the magnitude of the bias and not its spatial distribution. But it contains a lot of information about how the model is performing that is lost if we do e.g., a single Taylor diagram over 3D space. Clearly we are assuming that the reader has a certain level of familiarity with ocean circulation and the three-dimensional distribution of major tracers; we have tried to make it a bit more clear what we diagnose about model performance from these figures. We believe that Figures 2 and 3 give the reader adequate information to visualize the vertical distribution (although as the reviewer notes above, it excludes the abyssal depths).**

Ln. 444-450: how does this relate to any hard-wired limits in models?; the model I use, for instance, is prevented from consuming oxygen below a limit

**Our model does not have such a hardwired limit, but shifts respiration from O2 to NO3 below 6 uM.**

Ln. 461: re: "much deeper" - why?; and does this relate to the abyssal issue I raised re: oxygen?; i.e. this is an interesting depth

**We chose the depths for O2 because we believe that the processes that create and maintain the OMZs are of interest to readers. We chose a more abyssal depth for saturation because there is quite a lot of variability among models in terms of transporting DIC and alkalinity to the deep ocean.**

Figure 5: might a table be better for this information?; maybe combined with other measures of model performance?

**We think that the bar graph provides a compelling visual illustration of the key points we are trying to convey here.**

Figure 6: too many colours here makes it more difficult to discern differences between the panels

**Again, the ranges are large. In this case we used separate colour scales for the different depths in order to constrict the ranges as much as possible. We will reassess whether it is possible to shrink them further but the change is unlikely to be large.**

Figure 6: might the depth at which omega aragonite hits some threshold (value 1 would be most obvious) be better?

**Maps of the depth of the saturation horizon have been added to the Supplementary material.**

Ln. 471-472: again, remineralisation is mentioned in the context of biases before anything about production and export is introduced; omega is a downstream variable, so the ordering of the analysis here is perplexing

**This sentence has been deleted in accordance with the comments of Reviewer #1, who thought there was too much Discussion in the Results.**

Figure 7: how reliable are the observations here?; GLODAP is much less data-rich than WOA

**For the kind of global-scale analyses we are primarily concerned with here, the gridded data products are quite reliable. It is instructive to consider how little the gridded data product changed between GLODAP 1 and 2 (except in the Arctic of course). This can be in large part attributed to the foresight of the people who designed the first global survey (e.g., Feely et al., 2001, 10.5670/oceanog.2001.03).**

Figure 7: geographical plots of surface omega, seafloor omega, and the depth at which omega hits some threshold would seem more valuable to me; and easier to compare between models - these are very similar looking plots whose differences are not easy to discriminate

**We think these plots give the reader an indication of whether the models are doing a good job of representing the large-scale distribution of DIC and alkalinity, and their effect on the saturation state, which is a commonly used diagnostic of biological and geochemical impacts of anthropogenic CO2. We agree that there is a certain amount of redundancy here as OmegaA, OmegaC and [CO3--] are determined by similar processes. Note that in response to other reviewer comments we will be replacing these plots with ones that show only the observed distributions and the model anomalies relative to it, which probably brings out the differences a bit more starkly.**

**Maps of the depth of the saturation horizon have been added to the Supplementary material.**

Figure 7: do you need both aragonite and calcite?; the model seems to use calcite only

**The model assumes calcite for purposes of calculating burial/dissolution at the sediment/water interface, to avoid introducing unnecessary and unconstrained complexity. The saturation states of both minerals are determined by the distribution of DIC and alkalinity, regardless of what assumptions the model makes about the solid phases, and are of interest from the perspective of biological impacts and climate feedbacks.**

Figure 8: actually, I take my earlier comment back, the MEM is pretty much always better than the individual models

**See above and response to Reviewer #1.**

Ln. 496: as the N and Fe cycles regulate productivity and therefore ocean interior remineralisation and DIC/ALK, it would perhaps make more sense to discuss these ahead of the more downstream oxygen and carbonate chemistry properties

**This is addressed at the beginning in response to the reviewer's general comments.**

Ln. 502: HadGEM2-ES's marine BGC included a prognostic Fe cycle; see the full description of Totterdell (GMD, 2019)

**Yes we discovered this error on our own, after submission. Sorry about that. This text has been deleted.**

Figure 11a-11b: it seems overkill to have both 11a and 11b in the manuscript; I'd suggest deleting 11b

**Possibly we have a bit of a tropical bias, but we think that the seasonal cycle of equatorial upwelling and the associated HNLC condition is of interest to readers, and it is not readily discernible from Figure 11a. But more than one reviewer mentioned this, so Figure 11b has been moved to Supplementary.**

Figure 12: is this scale running across three orders of magnitude?; so is it 1 nmol/m3 to 1000 nmol/m3?; if so, the labelling of this log scale differs from that of the nitrate plots above

**We altered the colorbar of Figure 12 so that it is done in the same way as Figure 11 (actual data on a logarithmic scale rather than log(X) on a linear scale).**

Figure 14: why not geographical plots of DIN?

**These were included in an earlier draft and left out in the interest of space.**

Figure 18: worth plotting some regressions on here?; the data density means that the shape of the curves might be easier to discern then; also, why does the plot's chlorophyll appear "capped" at 1 mg / mg?

**Observational data > 1 mg m^-3 were excluded because the vast majority of these occur in coastal waters and are associated with processes not resolved by coarse resolution global models. In the open ocean, concentrations > 1 are very rare. This should have been stated in the caption and the Methods. This has been corrected.**

Figure 18: re: 17 mg / m3 - this seems a bad idea; why do this?; it looks like you're trying to maximise the appearance of fit

**The offset is clearly stated in both the caption and the legend, and the rationale for it is clearly explained in the text. Given the processes that are not considered at all in the model, its existence is unsurprising.**

Figure 19a: why crop the scale?; it's not helpful with bar charts

**We have replaced this plot with a version with the y axis starting at 0 as per the reviewer's suggestion.**

Ln. 666-673: a plot that might be helpful here is the geographical map of cumulative CO2 uptake; for instance, to identify whether the uptake pattern is the same but the magnitude different, or that there are actual differences in the spatial pattern of uptake

Ln. 666-673: another plot which might be useful here is the geographical inventory of anthro CO2 in the models (the CanESM5 ones); again to identify whether there are patterns in the differences between the models

**We added a Supplementary figure (S7) that shows column inventories of anthropogenic DIC for CanESM5 and CanESM5-CanOE. It does shed some light on the reasons for the difference in cumulative uptake between the two CanESM models, but the differences are not large.**

Ln. 681: is this decline in response to stratification happening here?; I thought the models were physically identical?

**The comment was simply that there has been a general consensus in the existing literature that the trend in global total export is likely to be downward in a warming ocean, but that the trends shown in the paper are difficult or impossible to verify using observations. The text of this paragraph has been rearranged to avoid confusion.**

Ln. 684: geographical plots of export production in the models, and how it changes between, say, 1980 and 2014 would be helpful here

**We showed zonal means because we thought the plot conveys the key message that the trend is fairly consistent across regions, and the processes differ between CMOC and CanOE in a somewhat consistent way, especially in the Southern Ocean. We could include difference maps as well, but we do not think they add much.**

Ln. 684: more generally, it seems strange to include these trends in export production without (a) talking about primary production, and (b) trying to dissect what the source(s) of the trends are

Figure 21: why focus on export ahead of production?; and would it be more interesting to consider export / production over this time period?

**We consider export production to be a more robust diagnostic of biological impacts on biogeochemical cycles than primary production. For example, a net change in export production will affect both ocean net $CO_2$ uptake and subsurface oxygen concentration, whereas a change in primary production does not necessarily affect either. We agree that the discussion of the underlying processes was superficial..**

Ln. 711-713: broken word here; "depend ... ent"

**fixed**

Ln. 768: are there runtime figures on how more costly it is?; you might expect cost to scale with complexity; e.g. Kwiatkowski et al. (2014) found ~linear relationships with tracer count

**Yes, as noted above the scaling is quite linear as discussed by Kwiatkowski et al.**

Ln. 798: my personal preference is to conclude a paper with a set of bulletpoint conclusions of the main findings

**The Discussion has been extensively restructured as per the comments of this reviewer and Reviewer #1. It is quite long and we think it ends in a appropriate fashion for this type of paper. The bulletpoints are a good idea generally but not really appropriate in this case.**

Ln. 807-809: maybe include the source ID for the model on the ESGF system together with the variant labels for the specific ensemble members included in the analysis

**This is now shown in Supplemental Table S3.**

Response to Reviewer# 3

Summary

The submitted manuscript by Christian and colleagues presents some major developments of the marine biogeochemistry component of the Canadian Earth System Model(s) v.5, focusing on representing a prognostic iron cycle and denitrification and including flexible phytoplankton elemental ratios and interactions between multiple food chains. These improvements are described in details and results of the Canadian Earth System model version (CanESM5-CanOE), which includes this newly improved marine biogeochemistry component, are presented and compared with results from two other CanESM versions (CanESM5- CMOC and CanESM2). While CanESM5-CMOC differs from CanESM5-CanOE in its ocean biogeochemistry component, CanESM2 is the older CanESM version, having different ocean circulation. The results show that CanESM5 versions are much better than CanESM2 when compared with available observations thanks to improvement in ocean circulation. The improvements in performance of CanESM5-CanOE over CanESM5- CMOC are not as clear due to sparsity in observations and uncertainties in historical trends. However, the inclusion of prognostic schemes for ocean Fe cycling and denitrification would be more suitable to address climate change problems.

Assessment

In general, I think that this manuscript is suitable for publication in Geoscientific Model Development, serving as a documentation on the development of an important model member of Earth System Models participating in CMIP. However, I do have some comments and suggestions, which hopefully can improve the quality of the manuscript.

First, while I understand that the main purpose of this manuscript is to describe recent developments in the ocean biogeochemistry component of the CanESM and to compare performance of its different versions, having more explanations as to why there are improvements of CanESM5-CanOE over CanESM5-CMOC in some areas but not all would be helpful. In addition, given that the comparison is performed also with CanESM2, which uses different ocean circulation, I would expect more discussions on which improvements of CanESM5 over CanESM2 are due to physics and which are due to biogeochemistry.

Second, I find the naming convention throughout the manuscript is somewhat confusing since there are three model versions are involved in the comparison, of which two are under the CanESM5 umbrella. Sometimes it is difficult to figure out which model version of the CanESM5 that the authors are referring to. In some places, the authors explicit wrote CanESM5-CanOE and CanESM5- CMOC, but in others, they wrote only CanESM5 or just CanOE and CMOC. It would be better if the authors could keep the naming consistent throughout the manuscript.

**We thank the reviewer for a thorough and constructive review. His concerns in the first two points overlap with those of the other reviewers, and we have addressed them in the revised MS.**

Third, since the model developments focus on Fe and N cycles, I was thinking that the authors should do a more comprehensive comparison of the modeled Fe distribution with observations, taking advantage of the growing GEOTRACES data. I understand that there is no climatological Fe dataset yet, but comparison with observed Fe transects from GEOTRACES should give an indication of the model performance on ocean Fe cycling.

**We added an additional Supplementary figure that compares modelled and observed concentrations along GA02 in the Atlantic, longest available transect. Mostly this confirms what we concluded about the model's biases from the existing analyses. There are a few novel points: (1) the model is biased high in the Antarctic Bottom Water, and (2) the seasonal biological drawdown in the mid-latitude North Atlantic is weak. Some discussion of possible reasons for these biases has been added to the text of the Supplementary information.**

Fourth, while export production is an important biogeochemistry feature, using it as a metric to evaluate model performance is difficult because of the uncertainty in the observational estimates, as the authors already pointed out. Primary production/chlorophyll might be a better metric.

**This issue was also raised by more than one reviewer, and the point is important. However, we believe our choice of plots and metrics is correct. Export production is not included mainly for purposes of model validation but rather, like $CO_2$ uptake, it is included so that readers can see how our models compare to other CMIP6 models on several global metrics that are commonly used and of broad interest.**

**We include several observation-based metrics of phytoplankton biomass (e.g., Figures 16-18) and present them in a way that we believe helps the reader understand the important differences in the way our two biology models are formulated. Aggregate export production is important for global ocean biogeochemistry and ocean $CO_2$ uptake; primary production is important for impacts on higher trophic levels but the same atoms can cycle faster or slower in the surface layer without any net uptake of CO2. Global spatial distribution of chlorophyll or primary production does not provide a very strong constraint on model performance due to the very strong enhancement in coastal regions that is unresolved by coarse resolution global models.**

Finally, since the historical trends section forms an important part of the manuscript, I would suggest the authors give more details on how the historical model runs are performed (i.e., which CO2 and atmospheric forcings are used…), how the results are analyzed, and why analyzing and comparing model historical trends is important.

**In accordance with the comments of this and other reviewers, we have provided a more detailed explanation for the inclusion of the historical trends section. While these are standard CMIP6 experiments, we have expanded the description of the experimental setup slightly as per the reviewer's suggestion, to make sure that there is no confusion or ambiguity.**

Some specific comments:

Line 27: some areas? Which areas? Please be more specific if possible.

**This is clarified in the revised MS.**

Line 30-32: Which CanESM version that shows these results?

**This is clarified in the revised MS.**

Line 127-128: Do you mean CanESM5 uses the same carbon chemistry as CanESM2?

**No. Carbon chemistry was slightly different in CanESM2, as the code was written before the current standard protocols were defined. This is clarified in the revised MS.**

Line 500: Change can not to cannot.

**Both of these are valid English. Possibly it is a difference between US and UK/Commonwealth English.**

Line 608-609: Which model version are you referring to here?

**This is clarified in the revised MS.**

Line 648-650: it might be worth to mention the difference between CanESM2 and CanESM5 in the nitrate initialization field earlier in the text. Introduction or section 2, for example.

**done**

---

## Author Response (AR2)

**The manuscript "Ocean biogeochemistry in the Canadian Earth System Model version 5.0.3: CanESM5 and CanESM5-CanOE" provides critical documentation for the CanESM5 and CanESM5-CanOE contributions to the 6th phase of the Coupled Model Intercomparison Project used in the IPCC 6th Assessment and as a general community research tool for studies in coupled carbon-climate change. The manuscript compares these models to the group's previous generation CMIP5, CanESM2, as well as the CMIP6 ensemble mean to demonstrate marked increase in skill with CanESM5-CanOE improving both biogeochemical skill while also increasing in comprehensiveness for interactive elemental cycles as a 'state of the art' model. As such, it is an important contribution to the scientific literature and should be published with minor revision to address technical points below.**

We thank the reviewer for a thorough and constructive review. As we understand it, this is a new reviewer who did not review the original submission. So in a few places we have provided context in terms of explaining why some things were changed or added in response to previous reviews.

**38-41 – The phasing of this sentence is confusing. I would shorten it to, "Cumulative ocean uptake of anthropogenic carbon dioxide through 2014 is lower in both CanESM5-CanOE (122 PgC) and CanESM5 (132 PgC) than in observation-based estimates (145 PgC) or the model ensemble mean (144 PgC)."**

done

**46 – It is not clear why the two Arora model application papers are being cited here in a statement about model development… a better reason to cite these papers would be in justifying an assertion that the Canadian models "have been contributing to coupled carbon-climate research" for over a decade and leave the citation of the necessary model development in support of these contributions to the Christian, et al., 2010 citation.**

Arora et al 2009 is expendable here, but Arora et al 2011 is not, as it contains key aspects of the CanESM2 model description. Possibly it is not really necessary to cite any of them at the end of this lead-in sentence, but it directs the reader to the most relevant publications for historical background. We deleted Arora et al 2009 and kept the other two.

**48 – CANESM5 includes not only a new ocean but updated atmosphere and land, as well… the atmospheric change could easily be more important than the ocean change.**

This is addressed in subsequent paragraphs (see below L94, 96). As this is an ocean-focused paper we chose to emphasize the ocean in the first paragraph of the Introduction. But actually the atmosphere did not change nearly as much as the ocean, and the differences in ocean circulation between CanESM2 and CanESM5 are mostly due to the adoption of the new ocean model.

**61 – need a comma after "7"**

done

**62 – The statement "2-3 times greater" seems very vague, computationally speaking. Is the variance between model computational costs really up to 50%? I would have expected a single number, e.g. "2.5 times greater, or much narrower range, e.g. "2.5-2.8 times greater"**

Yes this was a simplistic characterization based on the assumption that computation time scales linearly with the number of tracers, with some additional cost associated with the ocean model itself that is the same in each case. In practice it is ~2X. Wording changed to "CanOE is substantially more expensive computationally (19 tracers vs 7, so the total computation time to integrate the ocean model with biogeochemistry is approximately double)".

**87 – "CMOC N2 fixation" should be "that in CMOC" to avoid saying "Dinitrogen fixation"/"N2 fixation" twice.**

The small amount of extra text here is justified to avoid ambiguity; "that in" here could refer to N2 fixation rate, or to the model N2 fixation parameterization.

**88 – "there is no" should be "CMOC does not include"**

done

**94 – I would remove "ocean" as the authors are not attempting to distinguish which physical changes are due to the updated atmosphere model versus the updated ocean model, and the changes to the atmosphere could easily be more important than the changes to the ocean model.**

changed "ocean" to "climate"

**96 – Again, I would remove "ocean" as the Swart paper covers the full coupled model, not just the ocean.**

Changed to "An overall evaluation of the CanESM5 climate including the physical ocean is given in ..."

**144 – What is the evidence for this assertion? Is there a citation that could be provided and quantification, e.g. Orr et al., 2015 https://bg.copernicus.org/articles/12/1483/2015/**

**146-147 – Again, "This will affect the total ocean inventory of DIC but not the spatial distribution if the model is well equilibrated." Is an unsupported assertion… Is there evidence that the model is well equilibrated?**

We agree that both of the assertions are made without any reference to data, but these are fairly trivial points. If global total alkalinity is conserved and atmosphere CO2 is fixed, the model will converge on a total ocean inventory of DIC that is determined by the inventory of alkalinity. If the initial condition is a constant value or a fixed depth profile, it will take many years for the model to converge on its own internal equilibrium distribution of DIC, but this should not ultimately depend on the initial condition. That our models are relatively well equilibrated is demonstrated later in the paper (Table S6). To cite Orr et al 2015 here could be construed as misleading because the group of methods that they tested does not include the exact method employed in CanESM2. So the difference in pCO2 or Omega between the CanESM2 and

CanESM5 carbon chemistry could be a bit larger than it is in Orr et al, but it is still very small compared to the differences in the distribution of DIC between the two models.

**149 – While it is true that CanOE is a substantial advance over CMOS, the current wording of "The CanOE biology model is a substantially new model based on the cellular regulation model of Geider et al. (1998)." implies that CanOE is the first of its kind when in reality the equivalent IPSL, UK, NCAR, GFDL and other models have included this scope of ecosystem complexity intruded by Moore et al., 2004 and proliferated in the CMIP5 era… Rather, this statement should acknowledge that CanOE is not "substantially new", but rather an effort to meet the complexity that has become standard at other modeling centers. Suggest citing Seferain et al, (2020; https://link.springer.com/article/10.1007/s40641-020-00160-0) which documents the scope of BGC complexity in the CMIP6 class of models. The reader should understand that CanOE contains a level of BGC complexity akin to other models in CMIP6.**

"a substantially new model" was deleted (see also below 823-824)

**Table 1 – Many of the units are missing the element in which they are applied (e.g. kdnf is listed as "mmol m-3 d-1", but I think this should be "mmol N m-3 d-1", though there is some chance it might be "mmol C m-3 d-1").**

We reviewed the Table and the only parameters where there is potential ambiguity are K_DNF and K_Fe. The elemental symbols were added to the units of these.

**191 –The definition of "E" should have units… W m-2?**

units added

**287-289 – Given that CanOE represents N2 Fixation in a novel way that does not include PO4 limitation, it would be helpful to expand this discussion of N2 fixation with a couple of sentences and citations. In particular, "Dinitrogen fixation is parameterized as an external input of ammonium dependent on light, temperature and Fe availability, and inhibited by high ambient concentrations of inorganic N" should be augmented to point out how the maximum rate translates to a maximum vertically integrated rate given light availability much lower than typical recycled productivity in N-limited areas (I'm guessing something like ~0.5 mmol N m-2 d-1 for N2-fixation compared to ~3 mmol N m-2 d-1 for NH4 production) and the inhibition term serving to prevent runaway accumulation of fixed nitrogen. How were these scaling factors derived? Observations?, theory?, other models? An attempt to match denitrification and other terms to have a stable N-budget?, other?**

The parameterization derives partly from PISCES and partly from CMOC. This is made explicit in the revised text: "The temperature, iron and light limitation terms are based on PISCES (Aumont et al,, 2015); the N-inhibition term is from CMOC (Zahariev et al., 2008) (CMOC implicitly combines nitrate and ammonium into a single inorganic N pool)." We did not make much attempt to fine-tune the parameters because the global total N2 fixation falls within what we considered to be a reasonable range. The global integral is 68 TgN/y in the piControl, increasing to 82 Tg in 1995-2014.

0.5 mmol N m^-2 d^-1 is in the ballpark, although arguably on the high side. If we take HOT (22.75N, 158W) as a relatively well-characterized example, annual DNF was estimated as 40 mmolN m^-2 by Karl et al 1997 (Nature 388: 533). For a 40 m mixed layer with k_PAR=0.045 m^-1 and surface PAR of 150 W m^-2 (approximate summer value for HOT), the light-limitation function (Elim) will be ~0.75. If we assume T=25C, dFe=0.1 nM, and negligible DIN, the T and Fe limitation terms will each be around 0.5 and the N term ~1. So this gives a realized rate of ~0.15 mmol m^-2 d^-1 or an annual total of 54 mmolN m^-2. Of course, we also have to consider the seasonal and diel variation of surface PAR (time mean Elim not equal to Elim calculated for time-averaged irradiance) and its covariance with temperature. But this simple back-of-the envelope shows that the model produces rates in the expected range. The actual mean rate at this location in CanESM5-CanOE is 27 mmolN m^-2 y^-1 (mean of last 20 years of historical run).

**316-317 – The statement, "Burial in the sediments is represented as a simple 'on/off' switch dependent on the calcite saturation state (zero when $\Omega C<1$ and 1 when $\Omega C\geq1$)." It is unclear how the saturation state calculated given the statement on lines 139-140, "CanESM5 does not solve the carbon chemistry equations in the subsurface layers."… is carbon chemistry solved at the bottom? If so, lines 139-140 should be changed. If bottom saturation state is calculated some other way, this should be specified here.**

The reviewer is correct that in CanESM5 carbon chemistry is not solved in the subsurface layers, but in CanESM5-CanOE it is, as stated on 138-139. We added "In CanESM5-CanOE" at the beginning of this sentence to make sure there is no ambiguity.

**323 – comma needed after "production"**

**330 – comma needed before "but"**

both added

**339 – Is sediment burial of organic nitrogen considered?**

No. There is no burial of organic matter: all C and N in detritus reaching the sea floor are returned to the water column as DIC and NO3. We seem to have neglected to state this in the previous drafts, so we added "There is no burial of organic matter; organic matter reaching the seafloor is instantaneously remineralized." to section 2.5

**349 – "length scale of about 200 m"… shouldn't this length scale be precisely known rather than "about 200 m"**

No, because the parameterization is a bit more complex than a simple exponential decay (see Aumont et al 2015 eqs 85a-c); it approximates an exponential decay but the length scale is not exactly constant across the depth levels. The approximate e-folding length scale is actually closer to 600 m; this has been corrected.

**358-360 – Unless there are further constraints to be cited in the calibration, "particles available for it to precipitate onto, and assumes that POC is strongly positively correlated with total particulate matter." should just be "sinking particles" as equation 24 is not based**

on "POC" (which includes phytoplankton, bacteria, and some zooplankton) but rather only sinking detritus concentration, making any "assumption" between POC and total particulate matter only relevant insofar as there was some calibration performed which has not been referenced.

Added "detrital" before "POC". The parameterization is based on the concentration of suspended detritus (i.e., the nonliving fraction of POC). It is assumed that total particulate matter (including living biomass and mineral phases like $CaCO_3$ and biogenic silica) scales approximately linearly with Ds+Dl.

**373-374 – I interpret the statement "When the total fixed N adjustment is applied, one mole of alkalinity is removed per mole of N added or removed" that any positive and negative adjustments to the N budget are applied as a negative adjustment to alkalinity such that alkalinity will eventually be exhausted as the model runs to equilibrium… is that what was intended?**

This was incorrect; thanks for pointing this out. It has been changed to "one mole of alkalinity is added (removed) per mole of N removed (added)". This adjustment is necessary to maintain global conservation of alkalinity because nonphysically adding or removing nitrate short-circuits the overall pathway of N atoms through the 'fixed' pool (N2 fixation -> nitrification -> denitrification) which has a net alkalinity source/sink of 0 (Table S2).

**381-384 – Use of "inherently imperfect" in the statement "However, the OPA free surface formulation is inherently imperfect with regard to tracer conservation." Is vague… is this formulation globally conservative for mass, and/or Salinity, but not locally, or globally non-conservative? Is global non-conservation in mass or Salinity the source of drift for Alk and N, or is Salinity conserved? The statement "losses due to the free surface are generally larger for tracers with less homogeneous distributions" seems inconsistent with the drift being 3 times greater for surface Alk (range of about 25%; 2.2-2.7 eq m-3) versus NO3 (range of orders of magnitude; 0.00001-0.035 mol m-3)… my guess is that the non-conservation for salinity will be similar to that for Alk, and that the smaller effect for N is because much of the ocean has very low values relative to the global mean rather than a "homogeneous distribution".**

Yes the free surface is globally nonconservative, and this drift occurs in the salinity as well. The numbers were expressed as a fraction of the total tracer pool: 0.01%/ky for alkalinity and 0.03% for N. So it is ~3X larger for N. But it is true that the explanation for this discrepancy (the relative inhomogeneity of the surface distribution) is somewhat speculative and not something we have rigorously tested. So this parenthesis has been deleted.

**414/Figure 18 – "excluded"… were areas with Sat Chl >1 just set to 1, or really excluded from the comparison? The justification for excluding these observations as "mostly associated with coastal regions not resolved by coarse resolution global ocean models" is weak. A much better justification would be that these areas represent only a tiny fraction of the global ocean and would unduly weight any error calculation. It looks from Figure 18 that the model is able to represent Chl over 1 since these are provided, which makes this**

**artificial cutoff for the observations quite strange… at least one would expect the model and obs to both be capped at 1.**

Figure has been redrawn so that only data <1 mg m^-3 are shown for both model and observations.

**423 – Does "0.0011" have units? Is this the average anomaly between the 1x1 and 2x2 maps in mmol O2 m-3?**

No there are no units. It is the difference of pattern correlation coefficients calculated for a 1° or a 2° grid on the same depth level. The text has been revised slightly to make sure this is clear.

**427-428 – "as 20 year averages are used, internal variability is assumed to have little effect" would be more effectively put as "20 year averages are used to minimize expression of internal variability"**

done

**431 – need comma before "and"**

done

**446 – remove "here"**

done

**461 – "differences of" should be "and differences from that observational product for"**

done

**Figure 2, 3, and 6 – this figure should include statistics such as bias, r2 and std error or variance so that the reader can quantitatively compare the results. The r2 and relative variance are supplied graphically in the Taylor diagrams, so only the bias need be calculated anew.**

Average bias was added to each panel in Figures 2, 3, 6, and 7 (except OBS). Correlation is not really appropriate for anomalies. Standard deviation is arguably misleading to compare to the MEM, i.e., it is higher, but the same might be true for any individual model (as the reviewer notes, the pattern correlation and relative standard deviation are already shown for the depth levels in Figures 4+9). This argument could apply to bias as well (MEM cancels out opposing biases in various models), but we think the reviewer's suggestion to show average bias here is a good one. As expected, O2 and Omega are biased high except for Omega at 3500 m.

**531-532 – Looking at Figure 6 and Figure 7, the statement "CanESM5 and CanESM5-CanOE generally compare well with other models and observations." Appears true for 3500 m, but not at all true for 400 and 900 m in Figure 6 or the most of the water column in Figure 7 due to the strong high bias in saturation state that does not appear in the MEM. Of course comparing any single model to the ensemble mean is not a fair comparison. The statement should be revised to leverage quantitative information on model bias, r2, std error metrics. It is unfortunate that there is not more description of DIC and Alk biases to**

**help the reader understand why the saturation is biased so high. Is it just that the O2 is high and AOU is low? The patterns don't really look similar to me, so maybe not. Is the North Pacific ventilated too much? How does the model compare with natural 14C which was mentioned at the beginning of the manuscript?**

This is one of the pitfalls of showing only anomalies. The original submission showed the model values alongside the observed, with the anomalies (model minus observations) as Supplemental, but this was reversed in the first revision. The text states that the patterns "generally compare well" because when you look at the full modelled value the overall spatial pattern is similar to the observational data product or the MEM (Figure S2). The reviewer is correct that the positive bias is considerably larger in CanESM models than in the MEM, and that this is largely due to having a strongly ventilated mid-depth ocean (not only in the North Pacific, there is a substantial bias in the North Atlantic (Figure 6)). The text has been revised to include more information about the nature of the biases.

We have not yet published any dissi14c data. Looking at the "unofficial" data, the distribution is as expected: high in the mid-depth North Atlantic and low in the mid-depth North Pacific (e.g., 1000 m). One could make an argument that recently ventilated water penetrates deeper into the water column in CanESM5 than in the real world e.g. in the western North Pacific subtropical gyre (see e.g. Figure 2). But for this analysis to contribute significantly to understanding the underlying processes we would have to compare with a suite of other models, and there does not seem to be any DI14C in the GLODAP gridded data product.

**Caption to Figure 14 – Should "CMIP5" be "CMIP6"?**

Yes. Good catch. Thanks

**647 – Is "living" necessary here? Aren't all the phytoplankton groups "living"**

deleted

**720 – "CanESM models are biased low relative to observation based estimates"… is this true even after accounting for the missing mechanisms in an CMIP models as described in Bronselaer et al., 2017 (https://agupubs.onlinelibrary.wiley.com/doi/full/10.1002/2017GL074435) of lack of spinup at 1750 pCO2, 1750-1850 uptake, and legacy lack of uptake from the deltapCO2 between 1750 and 1850 leading to an additional 10-30 PgC( depending on integration period) not represented due to the experimental design? I would think the CanESMs would be pretty close to obs after accounting for these limitations of the CMIP experimental design. Also, is the saturation state high bias in these models also associated with a Revelle Factor bias? I would think it could lead to overly high CO2 uptake. It might be helpful to compare surface excess Alkalinity (Alk-DIC) to answer this.**

We thank the reviewer for pointing out this reference. We agree that some of the discrepancy between models and observations is due to it being in part an "apples and oranges" comparison. However, this affects other CMIP6 models as well, and both CanESM models are biased low not only relative to the observations but to other CMIP6 models that would share the same

corrections. The following sentence was added to the text: "Some of the difference may be attributable to differences in the way cumulative uptake is calculated in models vs observations (Bronselaer et al., 2017), although this should apply to other CMIP6 models as well."

We added a new Supplementary Figure to show zonal mean DIC-TA and added to the text "CanESM5 and CanESM5-CanOE show a high bias in near surface DIC relative to alkalinity (a measure of the ocean's capacity to absorb $CO_2$) in the mid-latitudes of both hemispheres (Supplementary Figure S8), which may in part explain the weak uptake of $CO_2$." If we draw global maps of DIC-TA at the surface they look very similar to the observations or the MEM, but there are slight biases, and the sign of the bias is consistent with the reviewer's hypothesis. The subsurface saturation state is a different question, and maybe not readily explicable in these terms. Possibly the cumulative CO2 uptake over 1850-2014 will depend on the bias in different depth strata and not just at the surface, but it is reasonable to assume that the biases in surface values calculated for e.g., 1986-2005 are fairly stable over time.

**785 – "the preindustrial control simulations had different degrees of equilibration when the historical experiment was launched (cf. Séférian et al., 2016, Supplementary Table S6)." The reader should not have to go to the supplemental of a model comparison paper to find the spin up length… however, when I went to Seferain Table 3, I see CanESM5 and CanESM5-CanOE as having been both spun up for 1000 years… so I am confused by this statement.**

Table S6 was added at the request of a reviewer of the previous draft, to put our comments about the drift fraction of DIC change in the historical run in context. It shows that the net ocean DIC change in the piControl is approximately 2X larger in CanESM5-CanOE than in CanESM5, but still small compared to anthDIC uptake. As discussed in Séférian et al. (2016), the spin-up was a bit ad hoc: possibly not as bad as in CMIP5, but not really ideal Best Practice either. We spun both models up ocean-only for as long as we could before launching the 1000 y coupled spinup, but the ocean-only spinup was longer for CanESM5 (partly because it runs faster and partly because it was ready sooner).

**815 – "It is also possible that the lower export production in CanESM5-CanOE is due to low iron supply to the surface waters of the Southern Ocean, but comparison with available observations do not suggest that this is the case." I don't see the logic for looking at iron biases here… Wouldn't it rather be a high bias in surface N and/or a low bias in the vertical N gradient that would suggest that the biological pup is too weak? The role of iron is really just to scale the degree of HNLC.**

Here we are trying to explain the rather large differences in global and especially Southern Ocean export production between CanESM2, CanESM5, and CanESM5-CanOE. Since only the latter has a prognostic iron cycle, it seems remiss not to discuss this. Since CanESM5-CanOE uses both an in-house parameterization of iron scavenging and a CanESM-derived aeolian Fe deposition field, it is possible that biases in one or both of these could explain the low EP; we think it is of interest to readers that this does not appear to be the case.

**823-824 – The statement "The development of CanOE was undertaken in response to some of the most severe limitations of CanESM2, and in light of our collective experience." Should indicate that these "limitations" were specifically with respect to biogeochemical and ecological comprehensiveness in other CMIP5 models such that CanOE would have "state-of-the-art" BGC for CMIP6 suitable for multi-hypothesis testing of interactions between elemental cycles in the climate change context.**

Added "Many of the additional features that CanOE introduces were already in the models published by other centres even in CMIP5." ("and in light of our collective experience" was deleted because with the new text it disrupts the flow.)

**877 – need comma after "achieved"**

done

**880 – "As to whether the gains in skill with CanESM5-CanOE justify the extra computational cost" – This simple expression of trade-off between tracer cost versus fidelity really undersells the achievement in CanOE as it ignores the difficulty of adding comprehensiveness and degrees of freedom without reducing fidelity (complex models are typically more difficult to optimize than simple models), the gain in robustness when mechanisms are more fully resolved (whether the model performance should be believed as "getting the right answer for the right reason"), and the gains in representing changing elemental interactions allowing the testing of new hypotheses. These are all achievements that should also be highlighted as justifying the extra computational cost.**

The first two points are actually addressed in the latter half of this paragraph: less parameterized models usually perform better in novel environments in part because they are more mechanistically based. We added a bit of additional text to emphasize this point more. We also appended to the end of the paragraph: "Inclusion of a prognostic iron cycle and C/N/Fe stoichiometry also open up additional applications and scientific investigations that are not possible with CMOC."

**899 – The phrase "and it is likely that the simplification of having a single particle sinking speed is not well suited to a domain with complex topography and prominent continental shelf and slope." Is very specific to a "domain" that has not been specified. As such the criticism, unless it is part of an established literature that is not currently cited, comes across as a non-sequitur pronouncing sentence before the crime has occurred and should be removed.**

Good point. This is the final paragraph and we were trying to communicate what we envision ourselves doing going forward and how it guides our choices about model structure. We changed it to: "We are also developing CanOE for regional downscaling applications (Hayashida, 2018; Holdsworth et al., 2021). The regional domains have complex topography and prominent continental shelf and slope, and the single remineralization length scale in CMOC may not be well suited to such an environment."